# Bootstrapping Upper Confidence Bound

**Botao Hao**
Purdue University
haobotao000@gmail.com

**Yasin Abbasi-Yadkori**
VinAI
yasin.abbasi@gmail.com

**Zheng Wen**
Deepmind
zhengwen@google.com

**Guang Cheng**
Purdue University
chengg@purdue.edu

## Abstract

Upper Confidence Bound (UCB) method is arguably the most celebrated one used in online decision making with partial information feedback. Existing techniques for constructing confidence bounds are typically built upon various concentration inequalities, which thus lead to over-exploration. In this paper, we propose a non-parametric and data-dependent UCB algorithm based on the multiplier bootstrap. To improve its finite sample performance, we further incorporate second-order correction into the above construction. In theory, we derive both problem-dependent and problem-independent regret bounds for multi-armed bandits with symmetric rewards under a much weaker tail assumption than the standard sub-Gaussianity. Numerical results demonstrate significant regret reductions by our method, in comparison with several baselines in a range of multi-armed and linear bandit problems.

## 1 Introduction

In artificial intelligence, learning to make decisions online plays a critical role in many fields, such as personalized news recommendation [1], robotics [2] and the game of Go [3]. To learn to make optimal decisions as soon as possible, the decision-makers must carefully design an algorithm to balance the trade-off between the exploration and exploitation [4, 5]. Over-exploration could be expensive and unethical in practice, e.g., medical decision making [6, 7, 8]. On the other hand, insufficient exploration tends to make an algorithm stuck at a sub-optimal solution. The delicate design of exploration methods stands in the heart of online learning and decision making.

Upper Confidence Bound (UCB) [9, 10, 11, 12, 13] is a class of highly effective algorithms in dealing with the exploration-exploitation trade-off in bandits and reinforcement learning. The tightness of confidence bound, as is known, is the key ingredient to achieve the optimal degree of explorations. To the best of our knowledge, nearly all the existing works construct confidence bounds based on various concentration inequalities, e.g. Hoeffding-type [10], empirical Bernstein type [14] or self-normalized type [13]. Those concentration-based confidence bounds, however, are typically conservative since they are *data-independent*. Concentration inequalities only exploit tail information, e.g., bounded or sub-Gaussian, rather than the whole distribution knowledge. In general, the loose constant factor may result in confidence bounds that are too wide to be informative [15].

In this paper, we propose a non-parametric and data-dependent UCB algorithm based on the multiplier bootstrap [16, 17, 18, 19, 20], called bootstrapped UCB. The principle is to use the multiplier

bootstrapped quantile as the confidence bound to enforce the exploration. Inspired by recent advances on non-asymptotic guarantee and non-asymptotic inference such as [18, 19, 20, 21], we develop an explicit second-order correction for the multiplier bootstrapped quantile that ensures the non-asymptotic validity. Our algorithm is easy to implement and has the potential to be generalized to more complicated models such as structured contextual bandits.

In theory, we develop both problem-dependent and problem-independent regret bounds for multi-armed bandits with symmetric rewards under a much weaker tail assumption, i.e., sub-Weibull distribution, than the classical sub-Gaussianity. In this case, it is proven that the mean estimator can still achieve the same problem-independent regret bound as the one under the sub-Gaussian assumption. Note that our result does not rely on other sophisticated approaches such as median-of-means or Catoni's M-estimator in [22]. A key technical tool we propose is a new concentration inequality for the sum of sub-Weibull random variables. Empirically, we evaluate our method in several multi-armed and linear bandit models. When the exact posterior is unavailable or the noise variance is mis-specified, the bootstrapped UCB demonstrates superior performance over variants of Thompson sampling and concentration-based UCB due to its non-parametric and data-dependent nature.

Recently, an increasing number of works [23, 24, 25, 26] study bootstrap methods for multi-armed and contextual bandits as an alternative to Thompson sampling. Most treat the bootstrap just as a way to randomize historical data (without any theoretical guarantee). One exception is [27] who derive a regret bound for Bernoulli bandit by adding pseudo observations. However, their method cannot be easily extended to unbounded cases, and their analyses heavily limit to the Bernoulli assumption. In contrast, our method applies to a broader class of bandit models with rigorous regret analysis.

The rest of the paper is organized as follows. Section 2 introduces the basic setup and our bootstrapped UCB algorithm. Section 3 provides the regret analysis and Section 4 conducts several experiments.

**Notations.** Throughout the paper, we denote $\mathbb{P}_{\boldsymbol{w}}(\cdot), \mathbb{E}_{\boldsymbol{w}}(\cdot)$ as the probability and expectation operator with respect to the distribution of the vector $\boldsymbol{w}$ only, conditioning on other random variables. We use similar notations for $\mathbb{P}_{\boldsymbol{y}}(\cdot), \mathbb{E}_{\boldsymbol{y}}(\cdot)$ with respect to $\boldsymbol{y}$ only. $[n]$ means the set $\{1, 2, \ldots, n\}$. We denote boldface lower letters (e.g. $\boldsymbol{x}, \boldsymbol{y}$) as a vector. For a set $\mathcal{E}$, we define its complement as $\mathcal{E}^c$.

## 2   Bootstrapped UCB

**Problem setup.** As a fruit fly, we illustrate our idea on the stochastic multi-armed bandit problem [28, 5]. In detail, the decision-makers interact with an environment for $T$ rounds. In round $t \in [T]$, the decision-makers pull an arm $I_t \in [K]$ and observes its reward $y_{I_t}$ which is drawn from a distribution associated with the arm $I_t$, denoted by $P_{I_t}$ with an unknown mean $\mu_{I_t}$. Without loss of generality, we assume arm 1 is the optimal arm, that is, $\mu_1 = \max_{k \in [K]} \mu_k$. In multi-armed bandit problems, the objective is to minimize the expected cumulative regret, defined as,

$$R(T) = T\mu_1 - \mathbb{E}\Big[\sum_{t=1}^{T} y_t\Big] = \sum_{k=2}^{K} \Delta_k \mathbb{E}\Big[\sum_{t=1}^{T} \mathbf{I}\{I_t = k\}\Big], \tag{2.1}$$

where $\Delta_k = \mu_1 - \mu_k$ is the sub-optimality gap for arm $k$, and $\mathbf{I}\{\cdot\}$ is an indicator function. Here, the second equality is from the regret decomposition Lemma (Lemma 4.5 in [5]). We call an upper bound of $R(T)$ problem-independent if the bound only depends on the distributional assumption and not on the specific bandit problem, say the gap $\Delta_k$.

**Upper Confidence Bound.** The upper confidence bound (UCB) algorithm [10] is based on the principle of optimism in the face of uncertainty. The key idea is to act as if the environment (parameterized by $\mu_k$ in multi-armed bandits) is as nice as plausibly possible. Concretely, a plausible environment refers to an upper confidence bound $\mathcal{G}(\boldsymbol{y}_n, 1 - \alpha)$ for the true mean $\mu$, of the form

$$\mathcal{G}(\boldsymbol{y}_n, 1 - \alpha) = \big\{x \in \mathbb{R}, x - \bar{y}_n \leq h_\alpha(\boldsymbol{y}_n)\big\}, \tag{2.2}$$

where $\boldsymbol{y}_n = (y_1, \ldots, y_n)^\top$ is the sample vector, $\bar{y}_n$ is the empirical mean, $\alpha \in (0, 1)$ is the confidence level, and $h_\alpha : \mathbb{R}^n \to \mathbb{R}^+$ is a threshold that could be either data-dependent or data-independent.

**Definition 2.1.** We define $\mathcal{G}(\boldsymbol{y}_n, 1 - \alpha)$ as a non-asymptotic upper confidence bound if for *any sample size* $n \geq 1$, the following inequality holds

$$\mathbb{P}\Big(\mu \in \mathcal{G}(\boldsymbol{y}_n, 1 - \alpha)\Big) \geq 1 - \alpha. \tag{2.3}$$

In bandit problems, a non-asymptotic control on the confidence level is more commonly used. This is rather different from the asymptotic validity of confidence bound in statistics literature [29].

A generic UCB algorithm will select the action based on its UCB index $\bar{y}_n + h_\alpha(\boldsymbol{y}_n)$ for different arms. As is well known, the sharper the threshold is, the better exploration and exploitation trade-off one can achieve [5]. By the definition of quantile, the sharpest threshold in (2.2) is the $(1-\alpha)$-quantile of the distribution of $\bar{y}_n - \mu$. However, this quantile relies on the knowledge of the exact reward distribution and is therefore itself unknown. To evaluate this value, we construct a data-dependent confidence bound based on the multiplier bootstrap.

## 2.1 Confidence Bound Based on Multiplier Bootstrap

**Multiplier Bootstrap.** Multiplier bootstrap is a fast and easy-to-implement alternative to the standard bootstrap, and has been successfully applied in various statistical contexts [18, 19, 20]. Its goal is to approximate the distribution of the target statistic by reweighing its summands with random multipliers independent of the data. For instance, in a mean estimation problem, we define a multiplier bootstrapped estimator as $n^{-1} \sum_{i=1}^n w_i(y_i - \bar{y}_n) = n^{-1} \sum_{i=1}^n (w_i - \bar{w}_n)y_i$, where $\{w_i\}_{i=1}^n$ are some random variables independent of $\boldsymbol{y}_n$, called bootstrap weights. Some classical weights are as follows:

- *Efron's bootstrap weights.* $(w_1, \ldots, w_n)$ is a multinomial random vector with parameters $(n; n^{-1}, \ldots, n^{-1})$. This is the standard nonparameteric bootstrap [30].
- *Gaussian weights.* $w_i$'s are i.i.d standard Gaussian random variables. This is closely related to Gaussian approximation in statistics [19].
- *Rademacher weights.* $w_i$'s are i.i.d Rademacher variables. This is closely related to symmetrization in learning theory.

The bootstrap principle suggests that the $(1-\alpha)$-quantile of the distribution of $n^{-1} \sum_{i=1}^n w_i(y_i - \bar{y}_n)$ conditionally on $\boldsymbol{y}_n$ could be used to approximate the $(1 - \alpha)$-quantile of the distribution of $\bar{y}_n - \mu$. As the first building block, the multiplier bootstrapped quantile is defined as,

$$q_\alpha(\boldsymbol{y}_n - \bar{y}_n) := \inf\left\{x \in \mathbb{R} \Big| \mathbb{P}_{\boldsymbol{w}}\Big(\frac{1}{n} \sum_{i=1}^n w_i(y_i - \bar{y}_n) > x\Big) \leq \alpha\right\}. \tag{2.4}$$

The question is whether $q_\alpha(\boldsymbol{y}_n - \bar{y}_n)$ is a valid threshold for any sample size $n \geq 1$.

## 2.2 Second-order Correction

Most statistical theories guarantee the *asymptotic validity* of $q_\alpha(\boldsymbol{y}_n - \bar{y}_n)$ by the multiplier central limit theorem [31]. However, we show that such a claim is valid *non-asymptotically* at the cost of adding a second-order correction. Next theorem rigorously characterizes this phenomenon under a symmetric assumption on the reward. Moreover, in Section A in the supplement, we show that without the second-order correction, a naive bootstrapped UCB will result in linear regret.

**Theorem 2.2** (Non-asymptotic Second-order Correction)**.** Suppose $\{y_i\}_{i=1}^n$ are i.i.d symmetric random variables with respect to its mean $\mu$, and the bootstrap weights $\{w_i\}_{i=1}^n$ are i.i.d Rademacher random variables. For two arbitrary parameters $\alpha, \delta \in (0, 1)$, the following inequality holds for any

sample size $n \geq 1$,

$$\mathbb{P}_{\boldsymbol{y}}\left(\bar{y}_n - \mu > \underbrace{q_{\alpha(1-\delta)}(\boldsymbol{y}_n - \bar{y}_n) + \sqrt{\frac{\log(2/\alpha\delta)}{n}}\varphi(\boldsymbol{y}_n)}_{\text{bootstrapped threshold}}\right) \leq 2\alpha, \qquad (2.5)$$

where $\varphi(\boldsymbol{y}_n)$ is a non-negative function satisfying $\mathbb{P}_{\boldsymbol{y}}(|\bar{y}_n - \mu| \geq \varphi(\boldsymbol{y}_n)) \leq \alpha$.

The detailed proof is deferred to Section B.1 in the supplement. In (2.5), the bootstrapped threshold may be interpreted as a main term, i.e., $q_{\alpha(1-\delta)}(\boldsymbol{y}_n - \bar{y}_n)$ (at a shrunk confidence level), plus a second-order correction term, i.e., $(\log(2/\alpha\delta)/n)^{1/2}\varphi(\boldsymbol{y}_n)$. The latter is added to guarantee the non-asymptotic validity of the bootstrapped threshold. In the above, $\varphi(\boldsymbol{y}_n)$ could be any preliminary upper bound on $\bar{y}_n - \mu$. Hence, Theorem 2.2 transforms a possibly coarse prior bound $\varphi(\boldsymbol{y}_n)$ on quantiles into a more accurate version that is based on a main term estimated by multiplier bootstrap plus a second-order correction term based on $\varphi(\boldsymbol{y}_n)$ multiplied by a $\mathcal{O}(n^{-1/2})$ factor.

**Remark 2.3** (Choice of $\varphi(\boldsymbol{y}_n)$)**.** If $\{y_i\}_{i=1}^n$ are independent 1-sub-Gaussian random variables, a natural choice of $\varphi(\boldsymbol{y}_n)$ is $(2\log(1/\alpha)/n)^{1/2}$ by Hoeffding's inequality (Lemma 2). Plugging it into (2.5) and letting $\delta = 1/2$, the bootstrapped threshold in (2.5) becomes

$$\underbrace{q_{\alpha/4}(\boldsymbol{y}_n - \bar{y}_n)}_{\text{main term}} + \underbrace{\frac{2\log(8/\alpha)}{n}}_{\text{second order correction}} . \qquad (2.6)$$

Lemma B.3 in the supplement shows that the main term is of order at least $\mathcal{O}(n^{-1/2})$ as $n$ grows, which implies the second order correction is just a remainder term. We emphasize that the reminder term is obviously not sharp and will be sharpened as a future work.

**Remark 2.4.** Existing works on UCB-type algorithms typically utilized various concentration inequalities, e.g. Hoeffding's inequality [10] or empirical Bernstein's inequality [14], to find a valid threshold $h_\alpha(\boldsymbol{y}_n)$. However, they are not data-dependent and only use the tail information, rather than fully exploit the whole distribution knowledge. This is typically conservative, and leads to over-exploration.

**Remark 2.5.** Empirical KL-UCB [32] used empirical likelihood to build confidence intervals for general distributions that have support in $[0, 1]$. Although empirical KL-UCB is also data-dependent, our proposed method is from a very different non-parametric perspective and uses different tools by bootstrap. In practice, resampling tends to be more efficient computationally, without solving a convex optimization each round like empirical KL-UCB. Moreover, our method can work with unbounded rewards and we believe it is easier to generalize to structured bandits, e.g. linear bandit.

In Figure 1, we compare different approaches to calculate 95% confidence bound for the population mean based on samples from a truncated-normal distribution. When the sample size is extremely small ($\leq 10$), the naive bootstrap (without any correction) cannot output a valid threshold since the bootstrapped quantile is smaller than the true 95% quantile. This confirms the necessity of the second-order correction. When the sample size increases, our bootstrapped threshold converges to the truth rapidly. This confirms the correction term is just a small remainder term. Additionally, the bootstrapped threshold is shown to be sharper than Hoeffding's bound and empirical Bernstein bound when sample size is large (see the right panel of Figure 1).

## 2.3 Main Algorithm: Bootstrapped UCB

Based on the above theoretical findings, we conclude that bootstrapped UCB will select the arm according to its UCB index defined as below:

$$\text{UCB}_k(t) = \bar{y}_{n_{k,t}} + q_{\alpha(1-\delta)}(\boldsymbol{y}_{n_{k,t}} - \bar{y}_{n_{k,t}}) + \sqrt{\frac{\log(2/\alpha\delta)}{n_{k,t}}}\varphi(\boldsymbol{y}_{n_{k,t}}), \qquad (2.7)$$

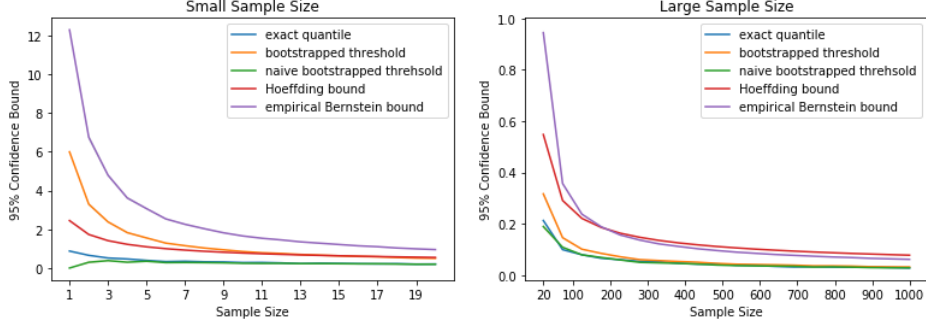

Figure 1: 95% confidence bound of the sample mean.

where $n_{k,t}$ is the number of pulls for arm $k$ until time $t$. Practically, we may use Monte Carlo quantile approximation to get an approximated bootstrapped quantile $\widetilde{q}_{\alpha(1-\delta)}(\boldsymbol{y}_{n_{k,t}} - \bar{y}_{n_{k,t}}, \boldsymbol{w}^B)$ and corresponding theorem for the control of the approximation of the bootstrapped quantile is also derived (see Section D in the supplement for details). The algorithm is summarized in Algorithm 1. The computational complexity at step $t$ is $\widetilde{\mathcal{O}}(Bt) \leq \widetilde{\mathcal{O}}(BT)$. Comparing with vanilla UCB, the extra $Bt$ is due to resampling. In practice, the choice of $B$ is seldom treated as a tuning parameter, but usually determined by the available computational resource.

---

**Algorithm 1** Bootstrapped UCB

---

**Input:** the number of bootstrap repetitions $B$, hyper-parameter $\delta$.
**for** $t = 1$ *to* $K$ **do**
  | Pull each arm once to initialize the algorithm.
**end**
**for** $t = K + 1$ *to* $T$ **do**
  | Set confidence level $\alpha = 1/(t+1)$.
  | Calculate the boostrapped quantile $\widetilde{q}_{\alpha(1-\delta)}(\boldsymbol{y}_{n_{k,t}} - \bar{y}_{n_{k,t}}, \boldsymbol{w}^B)$.
  | Pull the arm

$$I_t = \underset{k \in [K]}{\operatorname{argmax}}(\bar{y}_{n_{k,t}} + \widetilde{q}_{\alpha(1-\delta)}(\boldsymbol{y}_{n_{k,t}} - \bar{y}_{n_{k,t}}, \boldsymbol{w}^B) + (\log(2/\alpha\delta)/n_{k,t})^{1/2}\varphi(\boldsymbol{y}_{n_{k,t}})).$$

  | Receive reward $y_{I_t}$.
**end**

---

## 3 Regret Analysis

In Section 3.1, we derive regret bounds for bootstrapped UCB. Moreover, we show that naive bootstrapped UCB will result in linear regret in some cases in Section A in the supplement.

### 3.1 Regret Bound for Bootstrapped UCB

For multi-armed bandit problems, most literature [5] consider sub-Gaussian rewards. In this work, we move beyond sub-Gaussianity and consider the reward under a much weaker tail assumption, so-called sub-Weibull distribution. As shown in [33, 34], it is characterized by the right tail of the Weibull distribution and generalizes sub-Gaussian and sub-exponential distributions.

**Definition 1** (Sub-Weibull Distribution). We define $y$ as a sub-Weibull random variable if it has a bounded $\psi_\beta$-norm. The $\psi_\beta$-norm of $y$ for any $\beta > 0$ is defined as

$$\|y\|_{\psi_\beta} := \inf\left\{C \in (0, \infty) : \mathbb{E}[\exp(|y|^\beta/C^\beta)] \leq 2\right\}.$$

Particularly, when $\beta = 1$ or $2$, sub-Weibull random variables reduce to sub-exponential or sub-Gaussian random variables, respectively. It is obvious that the smaller $\beta$ is, the heavier tail the random variable has. Next theorem provides a corresponding concentration inequality for the sum of independent sub-Weibull random variables.

**Theorem 3.1** (Concentration Inequality for Sub-Weibull Distribution). Suppose $\{y_i\}_{i=1}^n$ are independent sub-Weibull random variables with $\|y_i\|_{\psi_\beta} \leq \sigma$. Then there exists an absolute constant $C_\beta$ only depending on $\beta$ such that for any $\boldsymbol{a} = (a_1, \ldots, a_n) \in \mathbb{R}^n$ and $0 < \alpha < 1/e^2$,

$$\Big| \sum_{i=1}^n a_i y_i - \mathbb{E}(\sum_{i=1}^n a_i y_i) \Big| \leq C_\beta \sigma \Big( \|\boldsymbol{a}\|_2 (\log \alpha^{-1})^{1/2} + \|\boldsymbol{a}\|_\infty (\log \alpha^{-1})^{1/\beta} \Big)$$

with probability at least $1 - \alpha$.

The proof relies on a precise characterization of $p$-th moment of a Weibull random variable and standard symmetrization arguments. Details are deferred to Section B.2 in the supplement. This theorem generalizes the Hoeffding-type concentration inequalities for sub-Gaussian random variables (see, e.g. Proposition 5.10 in [35]), and Bernstein-type concentration inequalities for sub-exponential random variables (see, e.g. Proposition 5.16 in [35]) up to some constants.

In Theorem 3.2, we provide both problem-dependent and problem-independent regret bounds.

**Theorem 3.2.** Consider a stochastic $K$-armed sub-Weibull bandit, where the noise follows a symmetric sub-Weibull distribution with its $\psi_\beta$-norm upper bounded by $\sigma$. Denote $n_{k,t}$ as the number of pulls for arm $k$ until time $t$. We choose $\varphi$ according to Theorem 3.1 as follows

$$\varphi(\boldsymbol{y}_{n_{k,t}}) = C_\beta \sigma \Big( \sqrt{\frac{\log 1/\alpha}{n_{k,t}}} + \frac{(\log 2/\alpha)^{1/\beta}}{n_{k,t}} \Big), \tag{3.1}$$

and let the confidence level $\alpha = 1/T^2$. For any round $T$, the problem-dependent regret of bootstrapped UCB is upper bounded by

$$R(T) \leq \sum_{k:\Delta_k > 0} 128 C_\beta^2 \sigma^2 \frac{\log T}{\Delta_k} + 2^{3+1/\beta} C_\beta \sigma K (\log T)^{1/\beta} + 4 \sum_{k=2}^K \Delta_k, \tag{3.2}$$

where $C_\beta$ is some absolute constant from Theorem 3.1, and $\Delta_k$ is the sub-optimality gap. Moreover, if the round $T \geq 2^{2/\beta - 3} K (\log T)^{2/\beta - 1}$, the problem-independent regret of bootstrapped UCB is upper bounded by

$$R(T) \leq 32\sqrt{2} C_\beta \sigma \sqrt{TK \log T} + 4K \mu_1^*. \tag{3.3}$$

The main proof structure follows the standard analysis of UCB [5] and relies on a sharp upper bound for the (data-dependent) bootstrapped quantile term by Theorem 3.1. Details are deferred to Section B.3 in the supplement. When $\beta \geq 1$, (3.2) provides a logarithm regret that matches the state-of-art result [5]. When $\beta < 1$, we have a non-negligible term $(\log T)^{1/\beta}$ that is the price paid for heavy-tailedness. However, this term does not depend on the gap $\Delta_k$. Therefore, we have an optimal problem-independent regret bound.

**Remark 3.3.** The choice of $\alpha = 1/T^2$ led to an easy analysis. Using similar techniques in Chapter 8.2 of [5], we can achieve a similar regret bound by setting $\alpha_t = 1/(t \log^\tau(t))$ for any $\tau > 0$.

**Remark 3.4.** [22] consider bandit with heavy-tail (moment of order $(1 + \varepsilon)$) based on a median-of-means estimator. As mentioned in [36], there are two disadvantages for median-of-means approach: (a) it involves an additional tuning parameter; (b) it is numerically unstable for small sample size. In contrast, we identify a class of heavy-tailed bandits (sub-Weibull bandit) where mean estimators can still achieve regret bounds of the same order as those under sub-Gaussian reward distributions. The reason is that although sub-Weibull r.v. has heavier tail than sub-Gaussian r.v., its tail still has an exponential-like decay.

# 4 Experiments

In Section 4.1, we consider multi-armed bandits with both symmetric and asymmetric rewards. In Section 4.2, we extend our method to linear bandits. Implementation details and some additional experimental results are deferred to Section E in the supplement.

## 4.1 Multi-armed Bandit

In this section, we compare bootstrapped UCB (Algorithm 1) with three baselines: Upper Confidence Bound based on concentration inequalities (Vanilla UCB), Thompson sampling with normal Jeffery prior [37] (Jeffery-TS) and Thompson sampling with Beta prior [38] (Bernoulli-TS). For bounded rewards, we also compare with Giro [27][1], that is a sampling-based exploration method by adding artificial pseudo observations $\{0, 1\}$ to escape from local optima, and empirical KL-UCB [39] using package: PymaBandits. For the preliminary bound $\varphi(\boldsymbol{y}_n)$, we simply choose the one derived by the concentration inequality. Note that the second-order correction term in (2.5) is conservative. For practitioners, we suggest to set the correction term to be $\varphi(\boldsymbol{y}_n)/\sqrt{n}$. To be fair, we choose the confidence level $\alpha = 1/(1 + t)$ for both UCB1 and bootstrapped UCB, and $\delta = 0.1$ in (2.5). All algorithms above require knowledge of an upper bound on the noise standard deviation. The number of bootstrap repetitions is $B = 200$, and the number of arms is $K = 5$.

First, we consider symmetric rewards with a mean parameter $\mu_k$ generated from Uniform$(-1, 1)$. The noise follows either truncated-normal distribution within $[-1, 1]$, or standard Gaussian distribution. From Figure 2, bootstrapped UCB outperforms Jeffery-TS and Vanilla-UCB for truncated-normal bandit and has comparable or sometimes better performance over empirical KL-UCB. It's obvious that if the reward distribution is exactly Gaussian and the plug-in estimate for the noise standard deviation is the truth, Jeffery-TS should be the best. However, when the posterior (plots (a),(b)) or noise standard derivation (plot (c)) are mis-specified, the performance of TS deteriorates fast. Since (concentration-based) Vanilla UCB only uses the tail information (bounded or sub-Gaussian), it is very conservative and results in bad regret as expected.

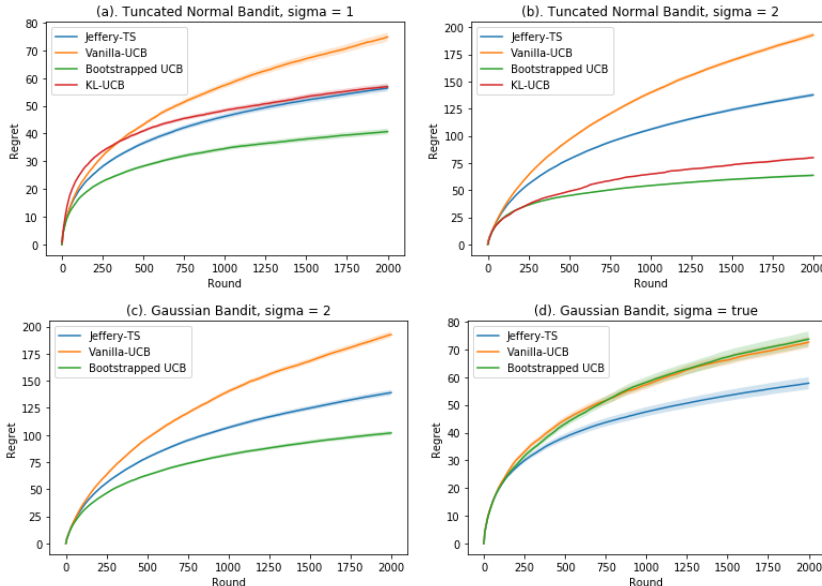

Figure 2: Cumulative regrets for truncated-normal bandit and Gaussian bandit. Sigma is the upper bound on the standard deviation of the noise. The results are averaged over 200 realizations.

Second, we consider asymmetric rewards with a mean parameter $\mu_k$ generated from Uniform$(0.25, 0.75)$. For Bernoulli bandit, the reward follows $\text{Ber}(\mu_k)$; for Beta bandit, the reward follows [2] $\text{Beta}(v\mu_k, v(1 - \mu_k))$ for $v = 8$. From Figure 3, bootstrapped UCB outperforms Vanilla UCB and Giro in both cases, and outperforms Bernoulli-TS for Beta bandit. In fact, we are supposed not to beat Bernoulli-TS for Bernoulli bandit since TS fully makes use of the distribution knowledge in this case. One possible explanation is that our method is non-parametric.

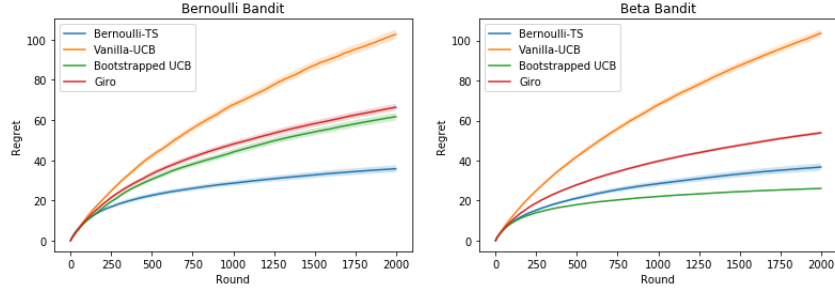

Figure 3: Cumulative regrets for Bernoulli bandit and Beta bandit. The results are averaged over 200 realizations.

Third, we demonstrate that the robustness of bootstrapped UCB over mis-specifications of the noise standard deviation. In the left panel of Figure 4, we consider the cumulative regret at round $T = 2000$ of standard Gaussian bandit. As one can see, when we increase the plug-in upper bound of the standard deviation of the noise, bootstrapped UCB is more robust than Bernoulli-TS and Vanilla UCB.

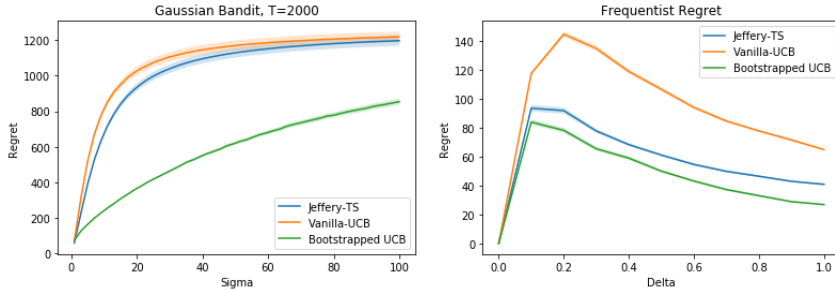

Figure 4: The left panel is the cumulative regret over noise levels while the right panel is the instance-dependent regret of various algorithms as a function of gaps. The results are averaged over 200 realizations.

Last, we present a frequentist instance-dependent regret curve for truncated-normal bandit and the experiment set up follows [40]. We plot cumulative regrets at $T = 1000$ of various algorithms with respect to the instance gap $\Delta$ and the mean vector $\mu = (\Delta, 0, 0, 0, 0)$. The results are summarized in the right panel of Figure 4.

## 4.2 Linear Bandit

We extend our method to linear bandit case. The basic set up follows the one in [15]. In detail, $\boldsymbol{\theta}^* \in \mathbb{R}^d$ is drawn from a multivariate Gaussian distribution with mean vector $\mu = 0$ and covariance matrix $\Sigma = 10I_d$. The noise follows a standard Gaussian distribution. There are 100 actions with feature vector components drawn uniformly at random from $[-1/\sqrt{10}, 1/\sqrt{10}]$. We consider two state-of-art methods: Thompson sampling for linear bandit [41] (TSL) and optimism in the face of uncertainty for linear bandits [13] (OFUL). Following the principle of constructing second-order

correction in mean problems (Theorem 2.2), we construct the bootstrapped UCB for linear bandit (BUCBL) as follows: At each round $t$, the action is selected as $\mathrm{argmax}_{\boldsymbol{x}}(\boldsymbol{x}^\top \widehat{\boldsymbol{\theta}}_t + \beta_{t,1-\delta}^{\mathrm{BUCBL}}\|\boldsymbol{x}\|_{V_t^{-1}})$, where $\beta_{t,1-\delta}^{\mathrm{BUCBL}} = q_\alpha(\widehat{\boldsymbol{\theta}}_t^{(b)} - \widehat{\boldsymbol{\theta}}_t) + \beta_{t,1-\delta,\sigma}^{\mathrm{OFUL}}/\sqrt{n}$. The formal definition of $\widehat{\boldsymbol{\theta}}_t, \widehat{\boldsymbol{\theta}}_t^{(b)}, \beta_{t,1-\delta,\sigma}^{\mathrm{OFUL}}$ and some basic setups are given in Section E.2 in the supplement. To be fair, the confidence level for all methods is set to be $\delta = 1/(1+t)$ and we plug in the true standard deviation of the noise for each method. From Figure 5, we can see that bootstrapped UCB greatly improves the cumulative regret over TSL and OFUL.

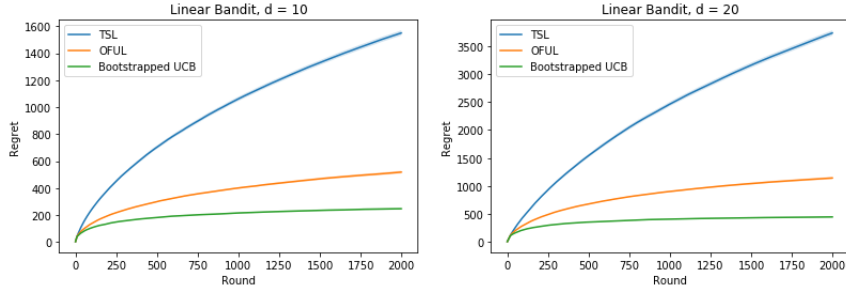

Figure 5: Cumulative regret for linear bandit.

# 5   Conclusion

In this paper, we propose a novel class of non-parametric and data-driven UCB algorithms based on multiplier bootstrap. It is easy to implement and has the potential to be generalized to other complex structured problems. As future works, we will evaluate our idea on other structured contextual bandits and reinforcement learning problems.

**Acknowledgments**

We thank Tor Lattimore for helpful discussions. Guang Cheng would like to acknowledge support by NSF DMS-1712907, DMS-1811812, DMS-1821183, and Office of Naval Research (ONR N00014-18-2759). In addition, Guang Cheng is a visiting member of Institute for Advanced Study, Princeton (funding provided by Eric and Wendy Schmidt) and visiting Fellow of SAMSI for the Deep Learning Program in the Fall of 2019; he would like to thank both Institutes for their hospitality.

## Footnotes

[1]We have implemented Giro in the unbounded reward case, which could result in linear regret in most cases. See Figure 7 in the supplement. So, it's unclear what is the best way to add pseudo observations in this case.

[2]We adopt the technique in [38] to run Thompson Sampling with $[0, 1]$ rewards. In particular, for any reward $y_t \in [0, 1]$, we draw pseudo reward $\widehat{y}_t \sim \text{Ber}(y_t)$, and then use $\widehat{y}_t$ instead of $y_t$ in the algorithm.

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
