[Supplementary Material]

# Supplement to "Bootstrapping Upper Confidence Bound"

In this supplement, we provide linear regret result in Section A, major proofs in Sections B and C. Some implementation details are in Sections D and E. In the end, we provide several supporting lemmas in Section F.

## A  Linear Regret

Following the augments in [42, 27], in this section, we show that UCB with a naive bootstrapped confidence bound will result in linear regret in two-armed Bernoulli bandit. At round $t+1$, the UCB index without the correction term for arm $k$ can be written as

$$\text{UCB}_k(t) = \bar{y}_{n_{k,t}} + q_{\alpha(1-\delta)}(\boldsymbol{y}_{n_{k,t}} - \bar{y}_{n_{k,t}}).$$

Consider the case where the first observation on the optimal arm is 0 but on the sub-optimal arm is 1. A key fact is that if the rewards are all zero, no matter how you bootstrap the data, the bootstrapped quantile is always zero. This will make the algorithm stuck into the sub-optimal arm.

**Theorem A.1.** Consider a stochastic 2-arm Bernoulli bandit with mean parameter $\mu_1, \mu_2$. The expected regret of the naive bootstrapped UCB can be lower bounded by

$$R(T) \geq \Delta_2 \Big( (1-\mu_1)\mu_2(T-2) + 1 \Big). \tag{A.1}$$

**Proof.**   Without loss of generality, we assume arm 1 is the optimal arm. Suppose at round $t = 1, 2$, we pull each arm once such that $y_1$ is with arm 1 and $y_2$ is with arm 2. Then we define a bad event as follows:

$$\mathcal{E} = \{y_1 = 0, y_2 = 1\}. \tag{A.2}$$

We know that under event $\mathcal{E}$, the decision-maker will never pull arm 1 any more starting from round $t = 3$. This is because if the rewards are all zero, no matter how you bootstrap the data, the bootstrapped quantile is always zero and then makes the decision-maker struck into the sub-optimal arm. Finally, we can lower bound the cumulative regret by,

$$
\begin{aligned}
R(T) &= \Delta_2 \mathbb{E}\Big[ \sum_{t=1}^{T} \mathbf{I}\{I_t = 2\} \Big] \\
&= \Delta_2 \mathbb{E}\Big[ \sum_{t=3}^{T} \mathbf{I}\{I_t = 2\}|\mathcal{E} \Big] \mathbb{P}(\mathcal{E}) + \Delta_2 \mathbb{E}\Big[ \sum_{t=3}^{T} \mathbf{I}\{I_t = 2\}|\mathcal{E}^c \Big] \mathbb{P}(\mathcal{E}^c) + \Delta_2 \\
&\geq \Delta_2 \mathbb{E}\Big[ \sum_{t=3}^{T} \mathbf{I}\{I_t = 2\}|\mathcal{E} \Big] \mathbb{P}(\mathcal{E}) + \Delta_2 \\
&= \Delta_2 T \mathbb{P}(y_1 = 0)\mathbb{P}(y_2 = 1) + \Delta_2 \\
&= \Delta_2 \Big( (1-\mu_1)\mu_2(T-2) + 1 \Big).
\end{aligned}
$$

This ends the proof.  ∎

We further demonstrate this phenomenon empirically for both Bernoulli bandit and Gaussian bandit in Figure 6.

## B  Proofs of Main Theorems

In this section, we provide detailed proofs of Theorems 2.2, 3.1 and 3.2.

### B.1  Proof of Theorem 2.2

The proof borrows the analysis from [18] but with refined analysis and sharp large deviation bound for binomial random variables.

Figure 6: Linear regret of naive bootstrapped UCB on Bernoulli bandit and Gaussian bandit. The result is averaged over 200 realizations.

**Step One.** Recall that (2.4) can be seen as the multiplier bootstrapped quantile around its empirical mean. We first takes advantage of the symmetry of each $\boldsymbol{y}$ around its mean by connecting the true quantile of $\bar{y}_n - \mu$ and the multiplier bootstrapped quantile around the true mean. Define the multiplier bootstrapped quantile around the true mean as

$$q_\alpha(\boldsymbol{y}_n - \mu) := \inf\left\{x \in \mathbb{R} \middle| \mathbb{P}_{\boldsymbol{w}}\left(\frac{1}{n}\sum_{i=1}^{n} w_i(y_i - \mu) > x\right) \le \alpha\right\}. \tag{B.1}$$

Since the probability operator $\mathbb{P}_{\boldsymbol{w}}$ is conditionally on $\boldsymbol{y}_n$, all the randomness of $q_\alpha(\boldsymbol{y}_n - \mu)$ come from $\boldsymbol{y}_n$. By the symmetric assumption of the reward, the distribution of $y_i - \mu$ is *exactly the same* as the distribution of $w_i(y_i - \mu)$ for Rademacher r.v. $\{w_i\}$. Then we have

$$\mathbb{P}\left(\bar{y}_n - \mu > q_\alpha(\boldsymbol{y}_n - \mu)\right)$$
$$= \mathbb{E}_{\boldsymbol{w}}\left[\mathbb{P}_{\boldsymbol{y}}\left(\frac{1}{n}\sum_{i=1}^{n} w_i(y_i - \mu) > q_\alpha((\boldsymbol{y}_n - \mu) \circ \boldsymbol{w}_n))\right)\right], \tag{B.2}$$

where $\circ$ is the Hadamard product. By Fubini's theorem, we can interchange the probability operator and expectation operator as follows

$$\mathbb{E}_{\boldsymbol{w}}\left[\mathbb{P}_{\boldsymbol{y}}\left(\frac{1}{n}\sum_{i=1}^{n} w_i(y_i - \mu) > q_\alpha((\boldsymbol{y}_n - \mu) \circ \boldsymbol{w}_n))\right)\right]$$
$$= \mathbb{E}_{\boldsymbol{y}}\left[\mathbb{P}_{\boldsymbol{w}}\left(\frac{1}{n}\sum_{i=1}^{n} w_i(y_i - \mu) > q_\alpha(\boldsymbol{y}_n - \mu)\right)\right] \le \alpha, \tag{B.3}$$

where the first inequality is due to the fact that for any arbitrary sign reversal, $q_\alpha((\boldsymbol{y}_n - \mu) \circ \boldsymbol{w}_n) = q_\alpha(\boldsymbol{y}_n - \mu)$ based on the definition of $q_\alpha$ and the last inequality is from the definition of quantitle. Combining (B.2) and (B.3) together, we conclude that

$$\mathbb{P}\left(\bar{y}_n - \mu > q_\alpha(\boldsymbol{y}_n - \mu)\right) \le \alpha. \tag{B.4}$$

**Step Two.** We define a good event

$$\mathcal{E} = \left\{\boldsymbol{y}_n \middle| q_\alpha(\boldsymbol{y}_n - \mu) \le q_{\alpha(1-\delta)}(\boldsymbol{y}_n - \bar{y}_n) + \sqrt{\frac{2\log(2/\alpha\delta)}{n}}\varphi(\boldsymbol{y}_n)\right\}. \tag{B.5}$$

Together with (B.4) and union event trick,

$$\mathbb{P}\Big(\bar{y}_n - \mu > q_{\alpha(1-\delta)}(\boldsymbol{y}_n - \bar{y}_n) + \sqrt{\frac{2\log(2/\alpha\delta)}{n}}\varphi(\boldsymbol{y}_n)\Big)$$

$$= \mathbb{P}\Big(\Big\{\bar{y}_n - \mu > q_{\alpha(1-\delta)}(\boldsymbol{y}_n - \bar{y}_n) + \sqrt{\frac{2\log(2/\alpha\delta)}{n}}\varphi(\boldsymbol{y}_n)\Big\} \cap \big(\{\boldsymbol{y}_n \in \mathcal{E}\} \cup \{\boldsymbol{y}_n \in \mathcal{E}^c\}\big)\Big)$$

$$= \mathbb{P}\Big(\Big\{\bar{y}_n - \mu > q_{\alpha(1-\delta)}(\boldsymbol{y}_n - \bar{y}_n) + \sqrt{\frac{2\log(2/\alpha\delta)}{n}}\varphi(\boldsymbol{y}_n)\Big\} \cap \{\boldsymbol{y}_n \in \mathcal{E}\}\Big)$$

$$+ \mathbb{P}\Big(\Big\{\bar{y}_n - \mu > q_{\alpha(1-\delta)}(\boldsymbol{y}_n - \bar{y}_n) + (2\log(2/\alpha\delta)/n)^{1/2}\varphi(\boldsymbol{y}_n)\Big\} \cap \{\boldsymbol{y}_n \in \mathcal{E}^c\}\Big)$$

$$\leq \mathbb{P}\Big(\bar{y}_n - \mu > q_{\alpha}(\boldsymbol{y}_n - \mu)\Big) + \mathbb{P}\Big(\boldsymbol{y}_n \in \mathcal{E}^c\Big)$$

$$\leq \alpha + \mathbb{P}(\boldsymbol{y}_n \in \mathcal{E}^c).$$

To bound $\mathbb{P}(\boldsymbol{y}_n \in \mathcal{E}^c)$, we first prove the following claim:

$$\text{Claim:} \quad \mathcal{E}^c \subset \Big\{\boldsymbol{y}_n | \mathbb{P}_{\boldsymbol{w}}\Big(\bar{w}_n(\bar{y}_n - \mu) > \sqrt{\frac{2\log(2/\alpha\delta)}{n}}\varphi(\boldsymbol{y}_n)\Big) \geq \alpha\delta\Big\}, \tag{B.6}$$

where $\bar{w}_n = \sum_{i=1}^n w_i/n$. To show this, we have by the definition of $q_\alpha(\boldsymbol{y}_n - \mu)$ in (B.1),

$$\mathbb{P}_{\boldsymbol{w}}\Big(\frac{1}{n}\sum_{i=1}^n w_i(y_i - \mu) > q_\alpha(\boldsymbol{y}_n - \mu)\Big) = \alpha.$$

By some simple algebras, we have

$$\frac{1}{n}\sum_{i=1}^n w_i(y_i - \mu) = \frac{1}{n}\sum_{i=1}^n w_i(y_i - \bar{y}_n + \bar{y}_n - \mu) = \frac{1}{n}\sum_{i=1}^n w_i(y_i - \bar{y}_n) + \bar{w}_n(\bar{y}_n - \mu). \tag{B.7}$$

For any $\boldsymbol{y}_n \in \mathcal{E}^c$,

$$\alpha = \mathbb{P}_{\boldsymbol{w}}\Big(\frac{1}{n}\sum_{i=1}^n w_i(y_i - \mu) > q_\alpha(\boldsymbol{y}_n - \mu)\Big)$$

$$\leq \mathbb{P}_{\boldsymbol{w}}\Big(\frac{1}{n}\sum_{i=1}^n w_i(y_i - \mu) > q_{\alpha(1-\delta)}(\boldsymbol{y}_n - \bar{y}_n) + \sqrt{\frac{2\log(2/\alpha\delta)}{n}}\varphi(\boldsymbol{y}_n)\Big) \text{ (by the definition of } \mathcal{E}^c\text{)}$$

$$= \mathbb{P}_{\boldsymbol{w}}\Big(\frac{1}{n}\sum_{i=1}^n w_i(y_i - \bar{y}_n) + \bar{w}_n(\bar{y}_n - \mu) > q_{\alpha(1-\delta)}(\boldsymbol{y}_n - \bar{y}_n) + \sqrt{\frac{2\log(2/\alpha\delta)}{n}}\varphi(\boldsymbol{y}_n)\Big) \text{ (by (B.7))}$$

$$\leq \mathbb{P}_{\boldsymbol{w}}\Big(\frac{1}{n}\sum_{i=1}^n w_i(y_i - \bar{y}_n) > q_{\alpha(1-\delta)}(\boldsymbol{y}_n - \bar{y}_n)\Big) + \mathbb{P}_{\boldsymbol{w}}\Big(\bar{w}_n(\bar{y}_n - \mu) > \sqrt{\frac{2\log(2/\alpha\delta)}{n}}\varphi(\boldsymbol{y}_n)\Big)$$

$$\leq \alpha(1-\delta) + \mathbb{P}_{\boldsymbol{w}}\Big(\bar{w}_n(\bar{y}_n - \mu) > \sqrt{\frac{2\log(2/\alpha\delta)}{n}}\varphi(\boldsymbol{y}_n)\Big).$$

This proves the claim of (B.6).

**Step Three.** We start to bound the second term above as follows,

$$\mathbb{P}_{\boldsymbol{w}}\Big(\bar{w}_n(\bar{y}_n - \mu) > \sqrt{\frac{2\log(2/\alpha\delta)}{n}}\varphi(\boldsymbol{y}_n)\Big) \tag{B.8}$$

$$\leq \mathbb{P}_{\boldsymbol{w}}\Big(|\bar{w}_n(\bar{y}_n - \mu)| > \sqrt{\frac{2\log(2/\alpha\delta)}{n}}\varphi(\boldsymbol{y}_n)\Big)$$

$$\leq \mathbb{P}_{\boldsymbol{w}}\Big(n|\bar{w}_n| > \sqrt{2n\log(2/\alpha\delta)}\frac{\varphi(\boldsymbol{y}_n)}{|\bar{y}_n - \mu|}\Big), \tag{B.9}$$

where the last inequality is actually conditional on the event $\{\bar{y}_n \neq \mu\}$ that holds with probability one. Note that $(w_i + 1/2) \sim \text{Bernoulli}(1/2)$ and thus $\sum_{i=1}^n (w_i + 1)/2 \sim \text{Binomial}(n, 1/2)$. Denote $X_n$

is a Binomial$(n, 1/2)$ random variable. Applying the sharp large deviation bound in Lemma 1 with $p_i = 1/2$, we have

$$
\begin{aligned}
\mathbb{P}_{X_n}\Big(X_n - \frac{n}{2} > \sqrt{2n\log(2/\alpha\delta)}\frac{\varphi(\boldsymbol{y}_n)}{|\bar{y}_n - \mu|}\Big) &\leq 2\exp\Big(-2\frac{\varphi(\boldsymbol{y}_n)^2}{(\bar{y}_n - \mu)^2}2n\log(2/\alpha\delta)\frac{1}{n}\Big) \\
&= 2\exp\Big(-\frac{4\log(2/\alpha\delta)\varphi(y_n)^2}{(\bar{\boldsymbol{y}}_n - \mu)^2}\Big). \quad \text{(B.10)}
\end{aligned}
$$

Putting (B.8) and (B.10) together,

$$
\mathbb{P}_{\boldsymbol{w}}\Big(\bar{w}_n(\bar{y}_n - \mu) > \sqrt{\frac{2\log(2/\alpha\delta)}{n}}\varphi(\boldsymbol{y}_n)\Big) \leq 2\exp\Big(-\frac{\log(2/\alpha\delta)\varphi(\boldsymbol{y}_n)^2}{(\bar{y}_n - \mu)^2}\Big).
$$

From (B.6), it remains to bound

$$
\begin{aligned}
\mathbb{P}\Big(\boldsymbol{y}_n \in \mathcal{E}^c\Big) &\leq \mathbb{P}_{\boldsymbol{y}}\Big(\mathbb{P}_{\boldsymbol{w}}\Big(\bar{w}_n(\bar{y}_n - \mu) > \sqrt{\frac{2\log(2/\alpha\delta)}{n}}\varphi(\boldsymbol{y}_n)\Big) \geq \alpha\delta\Big) \\
&\leq \mathbb{P}_{\boldsymbol{y}}\Big(2\exp\Big(-\frac{4\log(2/\alpha\delta)\varphi(\boldsymbol{y}_n)^2}{(\bar{y}_n - \mu)^2}\Big) \geq \alpha\delta\Big) \\
&= \mathbb{P}_{\boldsymbol{y}}\Big(|\bar{y}_n - \mu| \geq 2\varphi(\boldsymbol{y}_n)\Big).
\end{aligned}
$$

This reaches

$$
\mathbb{P}\Big(\bar{y}_n - \mu > q_{\alpha(1-\delta)}(\boldsymbol{y}_n - \bar{y}_n) + \sqrt{\frac{2\log(2/\alpha\delta)}{n}}\varphi(\boldsymbol{y}_n)\Big) \leq \alpha + \mathbb{P}_{\boldsymbol{y}_n}\Big(|\bar{y}_n - \mu| \geq \varphi(\boldsymbol{y}_n)\Big). \text{(B.11)}
$$

Letting $\varphi(\boldsymbol{y}_n)$ be a non-negative function such that

$$
\mathbb{P}_{\boldsymbol{y}}\Big(|\bar{y}_n - \mu| \geq \varphi(\boldsymbol{y}_n)\Big) \leq \alpha,
$$

we have

$$
\mathbb{P}\Big(\bar{y}_n - \mu > q_{\alpha(1-\delta)}(\boldsymbol{y}_n - \bar{y}_n) + \sqrt{\frac{2\log(2/\alpha\delta)}{n}}\varphi(\boldsymbol{y}_n)\Big) \leq 2\alpha.
$$

Redefine $\varphi(\boldsymbol{y}_n) = 2\varphi(\boldsymbol{y}_n)$ with a little bit abuse of notations. This ends our proof. ∎

## B.2   Proof of Theorem 3.1

We start by an upper bound for the $p$-th moment of sum of sub-Weibull random variables with bounded $\psi_\beta$-norm. The proof of Lemma B.1 is deferred to Section C.

**Lemma B.1.** Suppose $\{y_i\}_{i=1}^n$ are $n$ independent sub-Weibull random variables satisfying $\|y_i\|_{\psi_\beta} \leq \sigma$ with $\beta > 0$. Then for all $\boldsymbol{a} = (a_1, \ldots, a_n) \in \mathbb{R}^n$ and $p \geq 2$, we have

$$
\Big(\mathbb{E}\Big|\sum_{i=1}^n a_i y_i - \mathbb{E}(\sum_{i=1}^n a_i y_i)\Big|^p\Big)^{\frac{1}{p}} \leq \left\{ \begin{array}{ll} C_\beta\sigma\big(\sqrt{p}\|\boldsymbol{a}\|_2 + p^{1/\beta}\|\boldsymbol{a}\|_\infty\big), & \text{if } 0 < \beta < 1; \\ C_\beta\sigma\big(\sqrt{p}\|\boldsymbol{a}\|_2 + p^{1/\beta}\|\boldsymbol{a}\|_{\beta^*}\big), & \text{if } \beta \geq 1. \end{array} \right. \quad \text{(B.12)}
$$

where $1/\beta^* + 1/\beta = 1$, $C_\beta$ are some absolute constants only depending on $\beta$.

**Remark B.2.** If $0 < \beta < 1$, (B.12) is a combination of Theorem 6.2 in [43] and the fact that the $p$-th moment of a Weibull variable with parameter $\beta$ is of order $p^{1/\beta}$. If $\beta \geq 1$, (B.12) follows from a combination of Corollaries 2.9 and 2.10 in [44]. Continuing with standard symmetrization arguments, we reach the conclusion for general random variables. When $\beta = 1$ or 2, (B.12) coincides with standard moment bounds for a sum of sub-Gaussian and sub-exponential random variables in [35].

After we get the $p$-th moment bound in Lemma B.1, we can use Markov's inequality to transfer it to a high-probability as follows. For any $t > 0$, by Markov's inequality,

$$\mathbb{P}\Big( \Big| \sum_{i=1}^{n} a_i y_i - \mathbb{E}\Big( \sum_{i=1}^{n} a_i y_i \Big) \Big| \geq t \Big) = \mathbb{P}\Big( \Big| \sum_{i=1}^{n} a_i y_i - \mathbb{E}\Big( \sum_{i=1}^{n} a_i y_i \Big) \Big|^p \geq t^p \Big)$$

$$\leq \frac{\mathbb{E}\Big| \sum_{i=1}^{n} a_i y_i - \mathbb{E}\Big( \sum_{i=1}^{n} a_i y_i \Big) \Big|^p}{t^p} \leq \frac{C_\beta^p \sigma^p \Big( \sqrt{p} \|\boldsymbol{a}\|_2 + p^{1/\beta} \|\boldsymbol{a}\|_\infty \Big)^p}{t^p},$$

where the last inequality is from Lemma B.1. By setting $t$ such that

$$\exp(-p) = C_\beta^p \sigma^p (\sqrt{p} \|\boldsymbol{a}\|_2 + p^{1/\beta} \|\boldsymbol{a}\|_\infty)^p / t^p,$$

we have for $p \geq 2$,

$$\Big| \sum_{i=1}^{n} a_i y_i - \mathbb{E}\Big( \sum_{i=1}^{n} a_i y_i \Big) \Big| \leq e C_\beta \sigma \Big( \sqrt{p} \|\boldsymbol{a}\|_2 + p^{1/\beta} \|\boldsymbol{a}\|_\infty \Big)$$

holds with probability at least $1 - \exp(-p)$. Letting $\alpha = \exp(-p)$, we have that for any $0 < \alpha < 1/e^2$,

$$\Big| \sum_{i=1}^{n} a_i y_i - \mathbb{E}\Big( \sum_{i=1}^{n} a_i y_i \Big) \Big| \leq C_\beta \sigma \Big( \|\boldsymbol{a}\|_2 (\log \alpha^{-1})^{1/2} + \|\boldsymbol{a}\|_\infty (\log \alpha^{-1})^{1/\beta} \Big),$$

holds with probability at least $1 - \alpha$. This ends the proof. ∎

## B.3 Proof of Theorem 3.2

We first prove a problem-dependent bound then a problem-independent bound.

**Problem-Dependent Bound.** Recall that at round $t + 1$, the UCB index used in our algorithm is

$$\text{UCB}_k(t) = \bar{y}_{n_{k,t}} + h_\alpha(\boldsymbol{y}_{n_{k,t}}),$$

where $n_{k,t}$ is the number of pulls until round $t + 1$ for arm $k$ and

$$h_\alpha(\boldsymbol{y}_{n_{k,t}}) = q_{\alpha/2}\big(\boldsymbol{y}_{n_{k,t}} - \bar{y}_{n_{k,t}}\big) + \sqrt{\frac{2\log(4/\alpha)}{n_{k,t}}} \varphi(\boldsymbol{y}_{n_{k,t}}),$$

where

$$\varphi(\boldsymbol{y}_{n_{k,t}}) = C_\beta \sigma \Big( \sqrt{\frac{\log 1/\alpha}{n_{k,t}}} + \frac{(\log 2/\alpha)^{1/\beta}}{n_{k,t}} \Big). \tag{B.13}$$

From Theorem 3.1, for any fixed $n_{k,t} = s$, we know that

$$\mathbb{P}\Big( \bar{y}_s - \mu_k \geq \varphi(\boldsymbol{y}_s) \Big) \leq \alpha.$$

From Theorem 2.2, for any fixed $n_{k,t} = s$, we have

$$\mathbb{P}\Big( \mu_k - \bar{y}_s > h_\alpha(\boldsymbol{y}_s) \Big) \leq 2\alpha, \; k \in [K]. \tag{B.14}$$

The basic idea is to bound the expected number of pulls $\mathbb{E}(n_{k,t})$ for sub-optimal arms. To decouple the randomness from the behavior of the UCB algorithm, we define a good event as follows,

$$\mathcal{E}_k = \{\mu_1 < \min_{t \in [T]} \text{UCB}_1(t)\} \cap \{\bar{y}_{b_k} + h_\alpha(\boldsymbol{y}_{b_k}) < \mu_1\}, \; k \in [K], \tag{B.15}$$

where $b_k \in [T]$ is a constant to be chosen later.

First, we want to prove the following claim: if event $\mathcal{E}_k$ happens, then $n_{k,t} \leq b_k$. To show this, we use a contradiction argument. If $n_{k,t} > b_k$, then arm $k$ was pulled more than $b_k$ times over the first $T$ rounds, and so there must exist a round $t \in [T]$ such that $n_{k,t} = b_k$ and $I_t = k$. This implies

$$\text{UCB}_k(t) = \bar{y}_{n_{k,t}} + h_\alpha(\boldsymbol{y}_{n_{k,t}}) = \bar{y}_{b_k} + h_\alpha(\boldsymbol{y}_{b_k}).$$

From the definition of $\mathcal{E}_k$, we have

$$\bar{y}_{b_k} + h_\alpha(\boldsymbol{y}_{b_k}) < \mu_1 < \min_{t' \in [T]} \text{UCB}_1(t') \le \text{UCB}_1(t).$$

This results in a contradiction. Then we can decompose $\mathbb{E}[n_{k,t}]$ with respect to the event $\mathcal{E}_k$,

$$\mathbb{E}[n_{k,t}] = \mathbb{E}[I(\mathcal{E}_k)n_{k,t}] + \mathbb{E}[I(\mathcal{E}_k^c)n_{k,t}] \le b_k + \mathbb{P}(\mathcal{E}_k^c)T. \tag{B.16}$$

Second, we will derive an upper bound for $\mathbb{P}(\mathcal{E}_k^c)T$. By definition,

$$
\begin{aligned}
\mathbb{P}(\mathcal{E}_k^c) &= \mathbb{P}\Big( \{\mu_1 \ge \min_{t \in [T]} \text{UCB}_1(t)\} \cup \{\bar{y}_{b_k} + h_\alpha(\boldsymbol{y}_{b_k}) \ge \mu_1\} \Big) \\
&\le \underbrace{\mathbb{P}\Big(\mu_1 \ge \min_{t \in [T]} \text{UCB}_1(t)\Big)}_{I_1} + \underbrace{\mathbb{P}\Big(\bar{y}_{b_k} + h_\alpha(\boldsymbol{y}_{b_k}) \ge \mu_1\Big)}_{I_2}.
\end{aligned}
\tag{B.17}
$$

To bound $I_1$, we apply the union bound trick as follows,

$$
\begin{aligned}
\{\mu_1 \ge \min_{t \in [T]} \text{UCB}_1(t)\} &\subset \{\mu_1 \ge \min_{s \in [T]} \bar{y}_s + h_\alpha(\boldsymbol{y}_s)\} \\
&= \cup_{s \in [T]} \{\mu_1 \ge \bar{y}_s + h_\alpha(\boldsymbol{y}_s)\}.
\end{aligned}
$$

By B.14, it implies

$$\mathbb{P}\Big(\mu_1 \ge \min_{t \in [T]} \text{UCB}_1(t)\Big) \le \sum_{s=1}^{T} \mathbb{P}\Big(\mu_1 \ge \bar{y}_s + h_\alpha(\boldsymbol{y}_s)\Big) \le 2\alpha T. \tag{B.18}$$

To bound $I_2$, the key step is to derive an sharp upper bound for threshold $h_\alpha(\boldsymbol{y}_{b_k})$. Next lemma presents an upper bound for the multiplier bootstrapped quantile which is the main part of $h_\alpha(\boldsymbol{y}_{b_k})$. The proof is deferred to Section C.2.

**Lemma B.3.** Suppose $\{y_i - \mu\}_{i=1}^n$ follows sub-Weibull distribution with $\|y_i - \mu\|_{\psi_\beta} \le \sigma$ and $\{w_i\}_{i=1}^n$ are i.i.d Rademacher random variables independent of $y_i$. Then we have

$$\mathbb{P}\Big(\frac{1}{n}\sum_{i=1}^{n}(w_i - \bar{w})(y_i - \mu) \le C_\beta \sigma\Big(\sqrt{\frac{\log(1/\alpha)}{n}} + \frac{(\log(1/\alpha))^{1/\beta}}{n}\Big)\Big) \ge 1 - \alpha. \tag{B.19}$$

By the definition of $q_{\alpha/2}(\boldsymbol{y}_{b_k} - \bar{y}_{b_k})$ in (2.4), we have

$$q_{\alpha/2}(\boldsymbol{y}_{b_k} - \bar{y}_{b_k}) \le C_\beta \sigma\Big(\sqrt{\frac{\log(2/\alpha)}{b_k}} + \frac{(\log(2/\alpha))^{1/\beta}}{b_k}\Big), \tag{B.20}$$

with probability at least $1 - \alpha/2$. Recall that

$$\sqrt{\frac{2\log(4/\alpha)}{b_k}}\varphi(\boldsymbol{y}_{b_k}) = \sqrt{\frac{2\log(4/\alpha)}{b_k}}\Big(\sqrt{\frac{\log(1/\alpha)}{b_k}} + \frac{(\log(1/\alpha))^{1/\beta}}{b_k}\Big). \tag{B.21}$$

Overall, we have

$$
\begin{aligned}
h_\alpha(\boldsymbol{y}_{b_k}) &= q_{\alpha/2}(\boldsymbol{y}_{b_k} - \bar{y}_{b_k}) + \sqrt{\frac{2\log(4/\alpha)}{b_k}}\varphi(\boldsymbol{y}_{b_k}) \tag{B.22} \\
&\le 2C_\beta \sigma\Big(\sqrt{\frac{\log(2/\alpha)}{b_k}} + \frac{(\log(2/\alpha))^{1/\beta}}{b_k}\Big), \tag{B.23}
\end{aligned}
$$

with probability $1 - \alpha/2$ as long as $b_k \ge 2\log(4/\alpha)/(C_\beta^2 \sigma^2)$.

For two events $\mathcal{A}$ and $\mathcal{B}$, we have

$$\mathbb{P}(\mathcal{A}) = \mathbb{P}(\mathcal{A} \cap \mathcal{B}^c) + \mathbb{P}(\mathcal{A} \cap \mathcal{B}) \le \mathbb{P}(\mathcal{A} \cap \mathcal{B}) + \mathbb{P}(\mathcal{B}^c). \tag{B.24}$$

Next we define an event $\mathcal{B}_k = \{h_\alpha(\boldsymbol{y}_{b_k}) \leq \Delta_k/2\}$, where $\Delta_k = \mu_1 - \mu_k$. We decompose $I_2$ with respect to $\mathcal{B}_k$ following the union event rule (B.24),

$$
\mathbb{P}\Big( \bar{y}_{b_k} + h_\alpha(\boldsymbol{y}_{b_k}) \geq \mu_1 \Big)
$$

$$
= \quad \mathbb{P}\Big( \bar{y}_{b_k} + h_\alpha(\boldsymbol{y}_{b_k}) - \mu_k \geq \mu_1 - \mu_k \Big)
$$

$$
\leq \quad \mathbb{P}\Big( \bar{y}_{b_k} - \mu_k \geq \Delta_k - h_\alpha(\boldsymbol{y}_{b_k}) \cap \mathcal{B}_k \Big) + \mathbb{P}(\mathcal{B}_k^c)
$$

$$
\leq \quad \mathbb{P}\Big( \bar{y}_{b_k} - \mu_k \geq \frac{\Delta_k}{2} \cap \mathcal{B}_k \Big) + \mathbb{P}(\mathcal{B}_k^c)
$$

$$
\leq \quad \mathbb{P}\Big( \bar{y}_{b_k} - \mu_k \geq \frac{\Delta_k}{2} \Big) + \mathbb{P}(\mathcal{B}_k^c).
$$

To bound the first part, we reuse the concentration inequality in Theorem 3.1 such that,

$$
\mathbb{P}\Big( \bar{y}_{b_k} - \mu_k \geq \frac{\Delta}{2} \Big) \leq \exp\Big( -\min\Big[ \Big(\frac{\Delta_k}{C_\beta \sigma}\Big)^2 b_k, \Big(\frac{\Delta_k b_k}{4 C_\beta \sigma}\Big)^\beta \Big] \Big). \tag{B.25}
$$

To bound the second part, we bound $\mathbb{P}(\mathcal{B}_k^c)$ in three steps,

1. By (B.22), we have

$$
\mathbb{P}(\mathcal{B}_k^c) \quad = \quad \mathbb{P}\Big( h_\alpha(\boldsymbol{y}_{b_k}) > \Delta_k/2 \Big)
$$

$$
\leq \quad \mathbb{P}\Big( 2 C_\beta \sigma \Big( \sqrt{\frac{\log(2/\alpha)}{b_k}} + \frac{(\log(2/\alpha))^{1/\beta}}{b_k} \Big) > \Delta_k/2 \Big) + \alpha/2. \tag{B.26}
$$

2. To ensure that $2 C_\beta \sigma \sqrt{\log(2/\alpha)/b_k} \leq \Delta_k/4$, we need

$$
b_k \geq \Big( \frac{8 C_\beta \sigma}{\Delta_k} \Big)^2 \log(2/\alpha).
$$

To ensure that $2 C_\beta \sigma (\log(2/\alpha))^{(1/\beta)}/b_k \leq \Delta_k/4$, we need

$$
b_k \geq \frac{8 C_\beta \sigma (\log(2/\alpha))^{(1/\beta)}}{\Delta_k}.
$$

3. Then if we choose $b_k$ as

$$
b_k = \Big( \frac{8 C_\beta \sigma}{\Delta_k} \Big)^2 \log(2/\alpha) + \frac{8 C_\beta \sigma (\log(2/\alpha))^{1/\beta}}{\Delta_k}, \tag{B.27}
$$

we have

$$
\mathbb{P}\Big( 2 C_\beta \sigma \Big( \sqrt{\frac{\log(2/\alpha)}{b_k}} + \frac{(\log(2/\alpha))^{1/\beta}}{b_k} \Big) > \Delta_k/2 \Big) = 0. \tag{B.28}
$$

Combining (B.26) and (B.28), we conclude that when $b_k$ is choose as in (B.27), we have

$$
\mathbb{P}(\mathcal{B}_k^c) \leq \alpha/2. \tag{B.29}
$$

Combing (B.25) and (B.29), we have

$$
\mathbb{P}\Big( \bar{y}_{b_k} + h_\alpha(\boldsymbol{y}_{b_k}) \geq \mu_1 \Big) \leq \exp\Big( -\min\Big[ \Big(\frac{\Delta_k}{C_\beta \sigma}\Big)^2 b_k, \Big(\frac{\Delta_k b_k}{4 C_\beta \sigma}\Big)^\beta \Big] \Big) + \alpha/2, \tag{B.30}
$$

when $b_k$ is chosen as below

$$
b_k = \Big( \frac{8 C_\beta \sigma}{\Delta_k} \Big)^2 \log(1/\alpha) + \frac{8 C_\beta \sigma (\log(2/\alpha))^{1/\beta}}{\Delta_k}.
$$

Combining (B.17), (B.18) and (B.30) together,

$$
\begin{aligned}
\mathbb{P}(\mathcal{E}_k^c) &\leq 2T\alpha + \exp\Big(-\min\Big[\Big(\frac{\Delta_k}{C_\beta\sigma}\Big)^2 b_k, \Big(\frac{\Delta_k b_k}{4C_\beta\sigma}\Big)^\beta\Big]\Big) + \alpha/2 \\
&\leq 2T\alpha + \exp\Big(-\min\Big[\Big(\frac{\Delta_k}{C_\beta\sigma}\Big)^2\Big(\frac{8C_\beta\sigma}{\Delta_k}\Big)^2\log(2/\alpha), \Big(\frac{\Delta_k}{4C_\beta\sigma}\frac{8C_\beta\sigma(\log(2/\alpha))^{1/\beta}}{\Delta_k}\Big)^\beta\Big]\Big) + \alpha/2 \\
&= 2T\alpha + \exp\Big(-\min(64, 2^\beta)\log(2/\alpha)\Big) + \alpha/2.
\end{aligned}
\tag{B.31}
$$

Plugging (B.27), (B.31) into (B.16),

$$
\begin{aligned}
\mathbb{E}[n_{k,t}] &\leq b_k + \mathbb{P}(\mathcal{E}_k^c)T \\
&= \Big(\frac{8C_\beta\sigma}{\Delta_k}\Big)^2\log(2/\alpha) + \frac{8C_\beta\sigma(\log(2/\alpha))^{1/\beta}}{\Delta_k} + 2T^2\alpha + T\alpha^{\min(64,2^\beta)} + T\alpha/2.
\end{aligned}
$$

By choosing $\alpha = 2/T^2$, we have

$$
\mathbb{E}[n_{k,t}] \leq \Big(\frac{8C_\beta\sigma}{\Delta_k}\Big)^2 2\log T + \frac{8C_\beta\sigma}{\Delta_k}(2\log T)^{1/\beta} + 4,
\tag{B.32}
$$

since $1 - 2\min(64, 2^\beta) < 0$ for $\beta > 0$. Finally, the cumulative regret is upper bounded by

$$
\begin{aligned}
R(T) &= \sum_{k=2}^K \Delta_k \mathbb{E}[n_{k,t}] \tag{B.33} \\
&\leq \sum_{k=2}^K 128(C_\beta\sigma)^2\frac{\log T}{\Delta_k} + 8C_\beta\sigma K(2\log T)^{1/\beta} + 4\sum_{k=2}^K \Delta_k. \tag{B.34}
\end{aligned}
$$

This ends the proof.

**Problem-Independent Bound.** First we let $\Delta > 0$ as a threshold which will be specified later. Then we decompose $R(T)$ with respect to the value of $\Delta$ as follows,

$$
\begin{aligned}
R(T) &= \sum_{k=2}^K \Delta_k \mathbb{E}[n_{k,t}] \\
&= \sum_{k:\Delta_k < \Delta} \Delta_k \mathbb{E}[n_{k,t}] + \sum_{k:\Delta_k \geq \Delta} \Delta_k \mathbb{E}[n_{k,t}] \\
&\leq T\Delta + \sum_{k:\Delta_k \geq \Delta}\Big(128(C_\beta\sigma)^2\frac{\log T}{\Delta_k} + 8C_\beta\sigma(2\log T)^{1/\beta} + 4\Delta_k\Big) \\
&\leq 8C_\beta\sigma K(2\log T)^{1/\beta} + 4\sum_{k=2}^K \Delta_k + 128(C_\beta\sigma)^2\frac{K\log T}{\Delta} + T\Delta, \tag{B.35}
\end{aligned}
$$

where the first inequality is from (B.32). Letting $128(C_\beta\sigma)^2\frac{K\log T}{\Delta} = T\Delta$, we have

$$
\Delta = (128C_\beta^2\sigma^2\frac{K\log T}{T})^{1/2}.
\tag{B.36}
$$

Plugging (B.36) back into (B.35), we have

$$
R(T) \leq 2 * 128^{1/2}C_\beta\sigma\sqrt{TK\log T} + 4\sum_{k=1}^K \Delta_k + 8C_\beta\sigma K(2\log T)^{1/\beta}.
$$

When $T \geq 2^{2/\beta-3}K(\log T)^{2/\beta-1}$, we have

$$
R(T) \leq 32\sqrt{2}C_\beta\sigma\sqrt{TK\log T} + 4\sum_{k=2}^K \Delta_k \qquad \leq
$$

$$
32\sqrt{2}C_\beta\sigma\sqrt{TK\log T} + 4K\mu_1^*.
$$

This ends the proof. ∎

## C    Proofs of Main Lemmas

In this section, we provide the proofs of Lemmas B.1 and B.3.

### C.1    Proof of Lemma B.1

Without loss of generality, we assume $\|x_i\|_{\psi_\beta} = 1$ and $\mathbb{E}x_i = 0$ throughout this proof. Let $\beta = (\log 4)^{1/\beta}$. For notation simplicity, we define $\|x\|_p = (\mathbb{E}|x|^p)^{1/p}$ for a random variable $X$. The following step is to estimate the moment of linear combinations of variables $\{x_i\}_{i=1}^n$.

According to the symmetrization inequality (e.g., Proposition 6.3 of [45]), we have

$$\Big\| \sum_{i=1}^n a_i x_i \Big\|_p \leq 2 \Big\| \sum_{i=1}^n a_i \varepsilon_i x_i \Big\|_p = 2 \Big\| \sum_{i=1}^n a_i \varepsilon_i |x_i| \Big\|_p, \tag{C.1}$$

where $\{\varepsilon_i\}_{i=1}^n$ are independent Rademacher random variables and we notice that $\varepsilon_i x_i$ and $\varepsilon_i |x_i|$ are identically distributed. By triangle inequality,

$$
\begin{aligned}
2 \Big\| \sum_{i=1}^n a_i \varepsilon_i |x_i| \Big\|_p &\leq 2 \Big\| \sum_{i=1}^n a_i \varepsilon_i |x_i - \beta + \beta| \Big\|_p \\
&\leq 2 \Big\| \sum_{i=1}^n a_i \varepsilon_i |x_i - \beta| \Big\|_p + 2 \Big\| \sum_{i=1}^n a_i \varepsilon_i \beta \Big\|_p.
\end{aligned} \tag{C.2}
$$

Next, we will bound the second term of the RHS of (C.2). In particular, we will utilize Khinchin-Kahane inequality, whose formal statement is included in Lemma 5 for the sake of completeness. From Lemma 5 we have

$$
\begin{aligned}
\Big\| \sum_{i=1}^n a_i \varepsilon_i \beta \Big\|_p &\leq \Big( \frac{p-1}{2-1} \Big)^{1/2} \Big\| \sum_{i=1}^n a_i \varepsilon_i \beta \Big\|_2 \\
&\leq \beta \sqrt{p} \Big\| \sum_{i=1}^n a_i \varepsilon_i \Big\|_2.
\end{aligned} \tag{C.3}
$$

Since $\{\varepsilon_i\}_{i=1}^n$ are independent Rademacher random variables, some simple calculations implies

$$
\begin{aligned}
\Big( \mathbb{E} \Big( \sum_{i=1}^n \varepsilon_i a_i \Big)^2 \Big)^{1/2} &= \Big( \mathbb{E} \Big( \sum_{i=1}^n \varepsilon_i^2 a_i^2 + 2 \sum_{1 \leq i < j \leq n} \varepsilon_i \varepsilon_j a_i a_j \Big) \Big)^{1/2} \\
&= \Big( \sum_{i=1}^n a_i^2 \mathbb{E}\varepsilon_i^2 + 2 \sum_{1 \leq i < j \leq n} a_i a_j \mathbb{E}\varepsilon_i \mathbb{E}\varepsilon_j \Big)^{1/2} \\
&= \Big( \sum_{i=1}^n a_i^2 \Big)^{1/2} = \|\boldsymbol{a}\|_2.
\end{aligned} \tag{C.4}
$$

Combining inequalities (C.2)-(C.4),

$$2 \Big\| \sum_{i=1}^n a_i \varepsilon_i |x_i| \Big\|_p \leq 2 \Big\| \sum_{i=1}^n a_i \varepsilon_i |x_i - \beta| \Big\|_p + 2\beta \sqrt{p} \|\boldsymbol{a}\|_2. \tag{C.5}$$

Let $\{y_i\}_{i=1}^n$ be independent symmetric random variables satisfying $\mathbb{P}(|y_i| \geq t) = \exp(-t^\beta)$ for all $t \geq 0$. Then we have

$$
\begin{aligned}
\mathbb{P}(|x_i - \beta| \geq t) &\leq \mathbb{P}(x_i \geq t + \beta) + \mathbb{P}(x_i \leq \beta - t) \\
&\leq 2\mathbb{P}\big( \exp(|x_i|^\beta) \geq \exp((t+\beta)^\beta) \big) \\
&\leq 2(\mathbb{E}|x_i|^\beta) \cdot \exp(-(t+\beta)^\beta) \\
&\leq 4\exp(-(t+\beta)^\beta) \\
&\leq 4\exp(-t^\beta - \beta^\beta) = \mathbb{P}(|y_i| \geq t),
\end{aligned}
$$

which implies

$$\Big\|\sum_{i=1}^{n} a_i \varepsilon_i |x_i - \beta|\Big\|_p \le \Big\|\sum_{i=1}^{n} a_i \varepsilon_i y_i\Big\|_p = \Big\|\sum_{i=1}^{n} a_i y_i\Big\|_p, \tag{C.6}$$

since $\varepsilon_i y_i$ and $y_i$ have the same distribution due to symmetry. Combining (C.5) and (C.6) together, we reach

$$\Big\|\sum_{i=1}^{n} a_i x_i\Big\|_p \le 2\beta\sqrt{p}\|\boldsymbol{a}\|_2 + 2\Big\|\sum_{i=1}^{n} a_i y_i\Big\|_p. \tag{C.7}$$

For $0 < \beta < 1$, it follows Lemma 4 that

$$\Big\|\sum_{i=1}^{n} a_i y_i\Big\|_p \le C_\beta(\sqrt{p}\|\boldsymbol{a}\|_2 + p^{1/\beta}\|\boldsymbol{a}\|_\infty), \tag{C.8}$$

where $C_\beta$ is some absolute constant only depending on $\beta$.

For $\beta \ge 1$, we will combine Lemma 3 and the method of the integration by parts to pass from tail bound result to moment bound result. Recall that for every non-negative random variable $x$, integration by parts yields the identity

$$\mathbb{E}x = \int_0^\infty \mathbb{P}(x \ge t)dt.$$

Applying this to $x = |\sum_{i=1}^{n} a_i y_i|^p$ and changing the variable $t = t^p$, then we have

$$\begin{aligned}
\mathbb{E}|\sum_{i=1}^{n} a_i y_i|^p &= \int_0^\infty \mathbb{P}\Big(|\sum_{i=1}^{n} a_i y_i| \ge t\Big) p t^{p-1} dt \\
&\le \int_0^\infty 2\exp\Big(-c\min\Big(\frac{t^2}{\|\boldsymbol{a}\|_2^2}, \frac{t^\beta}{\|\boldsymbol{a}\|_{\beta^*}^\beta}\Big)\Big) p t^{p-1} dt,
\end{aligned} \tag{C.9}$$

where the inequality is from Lemma 3 for all $p \ge 2$ and $1/\beta + 1/\beta^* = 1$. In this following, we bound the integral in three steps:

1. If $\frac{t^2}{\|\boldsymbol{a}\|_2^2} \le \frac{t^\beta}{\|\boldsymbol{a}\|_{\beta^*}^\beta}$, (C.9) reduces to

$$\mathbb{E}|\sum_{i=1}^{n} a_i y_i|^p \le 2p \int_0^\infty \exp\Big(-c\frac{t^2}{\|\boldsymbol{a}\|_2^2}\Big) t^{p-1} dt.$$

Letting $t' = ct^2/\|\boldsymbol{a}\|_2^2$, we have

$$\begin{aligned}
2p \int_0^\infty \exp\Big(-c\frac{t^2}{\|\boldsymbol{a}\|_2^2}\Big) t^{p-1} dt &= \frac{p\|\boldsymbol{a}\|_2^p}{c^{p/2}} \int_0^\infty e^{-t'} t'^{p/2-1} dt' \\
&= \frac{p\|\boldsymbol{a}\|_2^p}{c^{p/2}} \Gamma\Big(\frac{p}{2}\Big) \le \frac{p\|\boldsymbol{a}\|_2^p}{c^{p/2}} \Big(\frac{p}{2}\Big)^{p/2},
\end{aligned}$$

where the second equation is from the density of Gamma random variable. Thus,

$$\Big(\mathbb{E}|\sum_{i=1}^{n} a_i y_i|^p\Big)^{\frac{1}{p}} \le \frac{p^{1/p}}{(2c)^{1/2}}\sqrt{p}\|\boldsymbol{a}\|_2 \le \frac{\sqrt{2}}{\sqrt{c}}\sqrt{p}\|\boldsymbol{a}\|_2. \tag{C.10}$$

2. If $\frac{t^2}{\|\boldsymbol{a}\|_2^2} > \frac{t^\beta}{\|\boldsymbol{a}\|_{\beta^*}^\beta}$, (C.9) reduces to

$$\mathbb{E}|\sum_{i=1}^{n} a_i y_i|^p \le 2p \int_0^\infty \exp\Big(-c\frac{t^\beta}{\|\boldsymbol{a}\|_{\beta^*}^\beta}\Big) t^{p-1} dt.$$

Letting $t' = ct^\beta / \|\boldsymbol{a}\|_{\beta^*}^\beta$, we have

$$2p \int_0^\infty \exp\Big(-c\frac{t^\beta}{\|\boldsymbol{a}\|_{\beta^*}^\beta}\Big)\Big)t^{p-1}dt \;=\; \frac{2p\|\boldsymbol{a}\|_{\beta^*}^p}{\beta c^{p/\beta}} \int_0^\infty e^{-t'} t'^{p/\beta-1} dt'$$

$$=\; \frac{2}{\beta}\frac{p\|\boldsymbol{a}\|_{\beta^*}^p}{c^{p/\beta}}\Gamma(\frac{p}{\beta}) \leq \frac{2}{\beta}\frac{p\|\boldsymbol{a}\|_{\beta^*}^p}{c^{p/\beta}}(\frac{p}{\beta})^{p/\beta}.$$

Thus,

$$\Big(\mathbb{E}|\sum_{i=1}^n a_i y_i|^p\Big)^{\frac{1}{p}} \leq \frac{2p^{1/p}}{(c\beta)^{1/\beta}}p^{1/\beta}\|\boldsymbol{a}\|_{\beta^*} \leq \frac{4}{(c\beta)^{1/\beta}}p^{1/\beta}\|\boldsymbol{a}\|_{\beta^*}. \qquad (C.11)$$

3. Overall, we have the following by combining (C.10) and (C.11),

$$\Big(\mathbb{E}|\sum_{i=1}^n a_i y_i|^p\Big)^{\frac{1}{p}} \leq \max\Big(\sqrt{\frac{2}{c}}, \frac{4}{(c\beta)^{1/\beta}}\Big)\Big(\sqrt{p}\|\boldsymbol{a}\|_2 + p^{1/\beta}\|\boldsymbol{a}\|_{\beta^*}\Big).$$

After denoting $C_\beta = \max\Big(\sqrt{\frac{2}{c}}, \frac{4}{(c\beta)^{1/\beta}}\Big)$, we reach

$$\Big\|\sum_{i=1}^n a_i y_i\Big\|_p \leq C_\beta\Big(\sqrt{p}\|\boldsymbol{a}\|_2 + p^{1/\beta}\|\boldsymbol{a}\|_{\beta^*}\Big). \qquad (C.12)$$

Since $0 < \beta < 1$, the conclusion can be reached by combining (C.7),(C.8) and (C.12). $\blacksquare$

## C.2  Proof of Lemma B.3

Note that with probability one,

$$\sum_{i=1}^n (w_i - \bar{w})^2 = \sum_{i=1}^n w_i^2 - n\bar{w} - n(1-\bar{w}) \leq n,$$
$$\max_i(w_i - \bar{w}) \leq 1.$$

We define a good event $\mathcal{E}$ as follows

$$\mathcal{E} = \Big\{\sum_{i=1}^n (w_i - \bar{w})^2 \leq n\Big\} \cup \Big\{\max_i(w_i - \bar{w}) \leq 1\Big\}. \qquad (C.13)$$

Then we decompose (B.19) conditional on $\mathcal{E}$,

$$\mathbb{P}\Big(\frac{1}{n}\sum_{i=1}^n (w_i - \bar{w})(y_i - \mu) \geq C_\beta\sigma\Big(\sqrt{\frac{\log 1/\alpha}{n}} + \frac{(\log 1/\alpha)^{1/\beta}}{n}\Big)$$

$$=\; \mathbb{P}\Big(\frac{1}{n}\sum_{i=1}^n (w_i - \bar{w})(y_i - \mu) \geq C_\beta\sigma\Big(\sqrt{\frac{\log 1/\alpha}{n}} + \frac{(\log 1/\alpha)^{1/\beta}}{n}\Big|\mathcal{E}\Big)\Big)\mathbb{P}(\mathcal{E})$$

$$+\mathbb{P}\Big(\frac{1}{n}\sum_{i=1}^n (w_i - \bar{w})(y_i - \mu) \geq C_\beta\sigma\Big(\sqrt{\frac{\log 1/\alpha}{n}} + \frac{(\log 1/\alpha)^{1/\beta}}{n}\Big|\mathcal{E}^c\Big)\Big)\mathbb{P}(\mathcal{E}^c)$$

$$\leq\; \mathbb{P}\Big(\frac{1}{n}\sum_{i=1}^n (w_i - \bar{w})(y_i - \mu) \geq C_\beta\sigma\Big(\sqrt{\frac{\log 1/\alpha}{n}} + \frac{(\log 1/\alpha)^{1/\beta}}{n}\Big|\mathcal{E}\Big)\Big)$$

$$\leq\; \mathbb{P}\Big(\frac{1}{n}\sum_{i=1}^n (w_i - \bar{w})(y_i - \mu) \geq C_\beta\sigma\Big(\frac{(\log 1/\alpha)^{1/2}}{n}\sqrt{\sum_{i=1}^n (w_i - \bar{w})^2} + \frac{(\log 1/\alpha)^{1/\beta}}{n}\max_i(w_i - \bar{w})\Big|\mathcal{E}\Big)\Big)$$

$$\leq\; \alpha,$$

where the first inequality is from $\mathbb{P}(\mathcal{E}^c) = 0$, the second inequality is from the independence of $w_i$ and $y_i$, the third inequality is from the concentration inequality in Theorem 3.1. This ends the proof. $\blacksquare$

## D   Monte Carlo Approximations

Suppose $n_{k,t}$ is the number of rewards associated with arm $k$ until round $t$. Practically, we could use Monte Carlo quantile approximation to calculate the multiplier bootstrapped quantile $q_\alpha(\boldsymbol{y}_{n_{k,t}} - \bar{y}_{n_{k,t}})$. Let $\{\boldsymbol{w}_n^{(1)}, \ldots, \boldsymbol{w}_n^{(B)}\}$ denote $B$ sets of independent random weight vectors and define

$$\widetilde{q}_\alpha(\boldsymbol{y}_n - \bar{y}_n, \boldsymbol{w}^B) := \inf\left\{ x \in \mathbb{R} \Big| \frac{1}{B}\sum_{b=1}^{B} \mathbf{I}\{\frac{1}{n}\sum_{i=1}^{n} w_i^{(b)}(y_i - \bar{y}_n) \geq x\} \leq \alpha \right\}, \qquad \text{(D.1)}$$

where $B$ is the number of bootstrap repetitions and $\boldsymbol{w}^B = (\boldsymbol{w}_n^{(1)}, \ldots, \boldsymbol{w}_n^{(B)})$. Then the UCB index for arm $k \in [K]$ can be written as

$$\text{UCB}_k(t) = \bar{y}_{n_{k,t}} + \widetilde{q}_{\alpha(1-\delta)}(\boldsymbol{y}_{n_{k,t}} - \bar{y}_{n_{k,t}}, \boldsymbol{w}^B) + \sqrt{\frac{2\log(2/\alpha\delta)}{n_{k,t}}}\varphi(\boldsymbol{y}_{n_{k,t}}). \qquad \text{(D.2)}$$

The decision-makers choose to pull arm $I_{t+1} = \text{argmax}_{k \in [K]} \text{UCB}_k(t)$. If $\text{UCB}_k(t) = \text{UCB}_{k'}(t)$ for $k \neq k'$, the tie is broken by a fixed rule that is chosen randomly in advance. Next theorem controls the approximation error of the bootstrapped quantile.

**Theorem D.1** (Monte Carlo Quantile Approximation). *Suppose the same conditions in Theorem 2.2 hold. We have*

$$\mathbb{P}_{\boldsymbol{y}, \boldsymbol{w}}(\bar{y}_n - \mu > \widetilde{q}_\alpha(\boldsymbol{y}_n - \bar{y}_n, \boldsymbol{w}^B) + \sqrt{\log(2/\alpha\delta)/n}\varphi(\boldsymbol{y}_n)) \leq \alpha + \frac{\lfloor B\alpha \rfloor + 1}{B+1} \leq 2\alpha + \frac{1}{B+1},$$

*where $\widetilde{q}_\alpha(\boldsymbol{y}_n - \bar{y}_n, \boldsymbol{w}^B)$ is the Monte Carlo approximated quantile defined in* (D.1).

By replacing the true quantile $q_\alpha$ by a MC quantile $\widetilde{q}_\alpha^B$ based on $B$ i.i.d bootstrapped weights, we lose at most $1/(B+1)$ for the confidence level.

*Proof Sketch.* The proof is similar to the proof of Theorem 2.2 except for the control of i.i.d approximation error. First, we define

$$\widetilde{q}_\alpha(\boldsymbol{y}_n - \mu, \boldsymbol{w}^B) := \inf\left\{ x \in \mathbb{R} \Big| \frac{1}{B}\sum_{b=1}^{B} \mathbf{I}\{\frac{1}{n}\sum_{i=1}^{n} w_i^{(b)}(y_i - \mu) \geq x\} \leq \alpha \right\}.$$

By using the similar symmetry properties as we did in (B.2) and (B.3), we have

$$\mathbb{E}_{\boldsymbol{w}^B}\mathbb{P}_{\boldsymbol{y}}\left(\frac{1}{n}\sum_{i=1}^{n} w_i(y_i - \mu) > \widetilde{q}_\alpha(\boldsymbol{y}_n - \mu, \boldsymbol{w}^B)\right)$$

$$= \mathbb{E}_{\boldsymbol{w}}\mathbb{E}_{\boldsymbol{w}^B}\mathbb{P}_{\boldsymbol{y}}\left(\frac{1}{n}\sum_{i=1}^{n} w_i(y_i - \mu) > \widetilde{q}_\alpha((\boldsymbol{y}_n - \mu) \circ \boldsymbol{w}_n, \boldsymbol{w}^B)\right)$$

$$= \mathbb{E}_{\boldsymbol{y}}\mathbb{P}_{\boldsymbol{w}, \boldsymbol{w}^B}\left(\frac{1}{n}\sum_{i=1}^{n} w_i(y_i - \mu) > \widetilde{q}_\alpha(\boldsymbol{y}_n - \mu, \boldsymbol{w}^B \cdot \text{diag}(\boldsymbol{w}_n))\right)$$

$$= \mathbb{E}_{\boldsymbol{y}}\mathbb{P}_{\boldsymbol{w}, \boldsymbol{w}^B}\left(\frac{1}{n}\sum_{i=1}^{n} w_i(y_i - \mu) > \widetilde{q}_\alpha(\boldsymbol{y}_n - \mu, \boldsymbol{w}^B)\right)$$

$$= \mathbb{E}_{\boldsymbol{y}}\mathbb{P}_{\boldsymbol{w}, \boldsymbol{w}^B}\left(\sum_{b=1}^{B} \mathbf{I}\{\frac{1}{n}\sum_{i=1}^{n} w_i^{(b)}(y_i - \mu) \geq x\} \leq \alpha\right) \leq \frac{\lfloor B\alpha \rfloor + 1}{B+1},$$

where the last inequality can be derived from Lemma 1 in [46]. The rest of the proof will follow step two in the proof of Section B.1. ∎

# E Additional Experimental Results and Implementation Details

In Section E.1, we present the implementation details for multi-armed bandits. In Section E.2, we present the implementation details for linear bandits. In Section E.3, we present formal definitions for logistic distribution and truncated-normal distribution.

## E.1 Multi-armd Bandit

For UCB1, at each round, the action is selected as

$$\operatorname*{argmax}_{k \in [K]} \frac{1}{n_k} \sum_{s=1}^{n_k} y_s^k + \widehat{\sigma} \sqrt{\frac{2 \log(1/\alpha)}{n_k}}.$$

For Jeffery-TS, at each round, the parameter is sampled from

$$\mathbb{N}\Big( \frac{1}{n_k} \sum_{s=1}^{n_k} y_s^k, \widehat{\sigma}^2 / n_k \Big).$$

Here, $\widehat{\sigma}$ is the upper bound on the estimator of standard deviation, $\{y_s^k\}$ are the reward associated with arm $k$ and $n_k$ is the number of reward associated with arm $k$. For notation simplicity, we ignore their dependency on round $t$.

In addition to Gaussian bandit and truncated-normal bandit, we also consider logistic bandit with parameter ($\mu = 0, s = 0.5$). The formal definition of logistic distribution and truncated-normal distribution. The results are summarized in Figure 7. Giro is almost failed.

Figure 7: Cumulative regret for logistic bandit. The left panel is for $\widehat{\sigma} = 1$, and the right panel is for $\widehat{\sigma} = 2$.

## E.2 Linear Bandit.

**Setup.** We particularly consider the following linear bandit setup. Let $\mathcal{D}_t \subset \mathbb{R}^d$ be an arbitrary (finite or infinite) set of arms. When an arm $\boldsymbol{x} \in \mathcal{D}_t$ is pulled, the agent receives a reward

$$y(\boldsymbol{x}) = \boldsymbol{x}^\top \boldsymbol{\theta}^* + \epsilon, \tag{E.1}$$

where $\boldsymbol{\theta}^* \in \mathbb{R}^d$ is the true reward parameter and $\epsilon$ is a zero-mean random noise with variance $\sigma^2$. We assume $\|\boldsymbol{\theta}^*\|_2 \leq S$. An arm $\boldsymbol{x} \in \mathcal{D}_t$ is evaluated according to its expected reward $\boldsymbol{x}^\top \boldsymbol{\theta}^*$ and for any $\boldsymbol{\theta} \in \mathbb{R}^d$, we denote the optimal arm and its value by

$$\boldsymbol{x}^*(\boldsymbol{\theta}) = \operatorname*{argmin}_{\boldsymbol{x} \in \mathcal{D}_t} \boldsymbol{x}^\top \boldsymbol{\theta}, \ J(\boldsymbol{\theta}) = \sup_{\boldsymbol{x} \in \mathcal{D}_t} \boldsymbol{x}^\top \boldsymbol{\theta}.$$

Thus $\boldsymbol{x}^* = \boldsymbol{x}^*(\boldsymbol{\theta}^*)$ is the optimal arm for $\boldsymbol{\theta}^*$ and $J(\boldsymbol{\theta}^*)$ is its optimal value. At each round $t$, the agent selects an arm $\boldsymbol{x}_t \in \mathcal{D}_t$ based on past observations. Then, it observes the reward $y_t = \boldsymbol{x}_t^\top \boldsymbol{\theta}^* + \epsilon_t$,

and it suffers a regret equal to the difference in expected reward between the optimal arm $\boldsymbol{x}^*$ and the arm $\boldsymbol{x}_t$. The objective of the agent is to minimize the cumulative regret up to round $t$,

$$R(T) = \sum_{t=1}^{T} \langle \boldsymbol{x}^* - \boldsymbol{x}_t, \boldsymbol{\theta}^* \rangle,$$

where $T$ is the time horizon. Note that the regret holds with high probability and thus is slightly from the standard notion of pseudo regret [13].

Denote $\boldsymbol{X}_t = (\boldsymbol{x}_1, \ldots, \boldsymbol{x}_t)^\top \in \mathbb{R}^{t \times d}$, $\boldsymbol{y}_t = (y_1, \ldots, y_t)^\top \in \mathbb{R}^{t \times 1}$. At round $t + 1$, consider a ridge estimator

$$\widehat{\boldsymbol{\theta}}_t = (\boldsymbol{X}_t^\top \boldsymbol{X}_t + \lambda \boldsymbol{I}_d)^{-1} \boldsymbol{X}_t \boldsymbol{y}_t. \tag{E.2}$$

Let us denote $V_t = \sum_{s=1}^{t} \boldsymbol{x}_s \boldsymbol{x}_s^\top \in \mathbb{R}^{d \times d}$ as the empirical covariance matrix.

**Algorithms.** For TSL: Thompson sampling for linear bandit [41], at each round $t$, the parameter is sampled as $\widetilde{\boldsymbol{\theta}}_t = \widehat{\boldsymbol{\theta}}_t + \widehat{\sigma}\sqrt{d \log(1/\delta)} V_t^{-1/2} \eta$ with $\eta \sim \mathbb{N}(0, I_d)$, where $\widehat{\sigma}$ is a standard deviation estimator. [41] suggests an even larger constant for the bonus term to enforce over exploration in theory. In practice, it will make the regret exploding. So we remove that large constant in our simulation.

For OFUL: optimism in the face of uncertainty for linear bandits [13], at each round $t$, the action is selected as $\operatorname{argmax}_{\boldsymbol{x}}(\boldsymbol{x}^\top \widehat{\boldsymbol{\theta}}_t + \beta_{t,1-\delta,\sigma}^{\text{OFUL}} \|\boldsymbol{x}\|_{V_t^{-1}})$, where

$$\beta_{t,1-\delta,\sigma}^{\text{OFUL}} = \widehat{\sigma}\sqrt{2 \log\left(\frac{\det(V_t)^{1/2} \det(\lambda \boldsymbol{I}_d)^{1/2}}{\delta}\right)} + \lambda^{1/2} S. \tag{E.3}$$

For BUCBL: bootstrapped UCB for linear bandit, we consider multinomial weights which is equivalent to sample with replacement. In detail, we generate $B$ sets of bootstrap repetitions $\{\boldsymbol{X}_t^{(b)}, \boldsymbol{y}_t^{(b)}\}$ from $\{\boldsymbol{X}_t, \boldsymbol{y}_t\}$ by sample with replacement, and calculate corresponding bootstrapped estimator

$$\widehat{\boldsymbol{\theta}}_t^{(b)} = (\boldsymbol{X}_t^{(b)\top} \boldsymbol{X}_t^{(b)} + \lambda \boldsymbol{I}_d)^{-1} \boldsymbol{X}_t^{(b)} \boldsymbol{y}_t^{(b)}, \tag{E.4}$$

and $V_t^{(b)} = \sum_{s=1}^{t} \boldsymbol{x}_s^{(b)} \boldsymbol{x}_s^{(b)\top}$. Define the bootstrapped weighted $\ell_2$-norm as follow

$$\|\widehat{\boldsymbol{\theta}}_t^{(b)} - \widehat{\boldsymbol{\theta}}_t\|_{V_t^{(b)} + \lambda \boldsymbol{I}_d} = \sqrt{(\widehat{\boldsymbol{\theta}}_t^{(b)} - \widehat{\boldsymbol{\theta}}_t)^\top (V_t^{(b)} + \lambda \boldsymbol{I}_d)(\widehat{\boldsymbol{\theta}}_t^{(b)} - \widehat{\boldsymbol{\theta}}_t)}.$$

For each set of bootstrap repetitions, we could calculate the $\|\widehat{\boldsymbol{\theta}}_t^{(b)} - \widehat{\boldsymbol{\theta}}_t\|_{V_t^{(b)} + \lambda \boldsymbol{I}_d}$ accordingly. Therefore, the bootstrapped threshold is defined as

$$q_\alpha(\widehat{\boldsymbol{\theta}}_t^{(b)} - \widehat{\boldsymbol{\theta}}_t) := (1 - \alpha)\text{-quantile of } \left\{ \|\widehat{\boldsymbol{\theta}}_t^{(1)} - \widehat{\boldsymbol{\theta}}_t\|_{V_t^{(1)} + \lambda \boldsymbol{I}_d}, \ldots, \|\widehat{\boldsymbol{\theta}}_t^{(B)} - \widehat{\boldsymbol{\theta}}_t\|_{V_t^{(B)} + \lambda \boldsymbol{I}_d} \right\}. \tag{E.5}$$

At each round $t$, the action is selected as $\operatorname{argmax}_{\boldsymbol{x}}(\boldsymbol{x}^\top \widehat{\boldsymbol{\theta}}_t + (q_\alpha(\widehat{\boldsymbol{\theta}}_t^{(b)} - \widehat{\boldsymbol{\theta}}_t) + \beta_{t,1-\delta,\sigma}^{\text{OFUL}}/\sqrt{n})\|\boldsymbol{x}\|_{V_t^{-1}})$.

### E.3 Logistic Distribution and Truncated-Normal Distribution

**Logistic Distribution** In probability theory and statistics, the logistic distribution is a continuous probability distribution. Its cumulative distribution function is the logistic function, which appears in logistic regression and feed forward neural networks. It resembles the normal distribution in shape but has heavier tails.

**Definition E.1.** The probability density function (pdf) of the logistic distribution $(\mu, s)$ is given by:

$$f(x) = \frac{\exp(-(x - \mu)/s)}{s(1 + \exp(-(x - \mu)/s))^2},$$

where $\mu$ is a location parameter and $s > 0$ is a scale parameter. The mean is $\mu$ and the variance is $s^2 \pi^2 / 3$.

**Truncated-normal Distribution**   In probability and statistics, the truncated normal distribution is the probability distribution derived from that of a normally distributed random variable by bounding the random variable from either below or above (or both).

**Definition E.2.** Suppose $X$ has a normal distribution with mean $\mu$ and variance $\sigma^2$ and lies within the interval $(a, b)$. Then $X$ conditional on $a < X < b$ has a truncated normal distribution $(\mu, a, b)$. Its probability density function $f$ is given by

$$f(x) = \frac{\phi(\frac{x-\mu}{\sigma})}{\sigma(\Phi(\frac{b-\mu}{\sigma}) - \Phi(\frac{a-\mu}{\sigma}))},$$

where $\phi(\cdot)$ is the probability density function of the standard normal distribution and $\Phi(\cdot)$ is its cumulative distribution function.

## F   Supporting Lemmas

**Lemma 1** (Large Deviation Bound, Theorem A.1.4 in [47]). Suppose $x_1, \ldots, x_n$ are mutually independent random variables with distribution

$$\mathbb{P}(x_i = 1 - p_i) = p_i, \ \mathbb{P}(x_i = -p_i) = 1 - p_i,$$

where $p_i \in [0, 1]$. For any $a > 0$, we have

$$\mathbb{P}\Big( \sum_{i=1}^{n} x_i > a \Big) < \exp(-2a^2/n).$$

When all $p_i = p$, the sum $\sum_{i=1}^{n} X_i$ has distribution Binomial$(n, p) - np$ where $B(n, p)$ is the Binomial distribution.

**Lemma 2** (Hoeffding's inequality, Proposition 5.10 in [35]). Let $X_1, \ldots, X_n$ be independent centered sub-Gaussian random variables, and let $K = \max_i \|X_i\|_{\phi_2}$. Then for any $\boldsymbol{a} = (a_1, \ldots, a_n)^\top$ and any $t > 0$, we have

$$\mathbb{P}\Big( |\sum_{i=1}^{n} a_i X_i| > t \Big) \leq e \exp\Big( - \frac{ct^2}{K^2 \|\boldsymbol{a}\|_2^2} \Big).$$

**Lemma 3** (Tail Probability for the Sum of Weibull Distributions (Lemma 3.6 in [48])). Let $\alpha \in [1, 2]$ and $Y_1, \ldots, Y_n$ be independent symmetric random variables satisfying $\mathbb{P}(|Y_i| \geq t) = \exp(-t^\alpha)$. Then for every vector $\boldsymbol{a} = (a_1, \ldots, a_n) \in \mathbb{R}^n$ and every $t \geq 0$,

$$\mathbb{P}\Big( |\sum_{i=1}^{n} a_i Y_i| \geq t \Big) \leq 2 \exp\Big( - c \min\Big( \frac{t^2}{\|\boldsymbol{a}\|_2^2}, \frac{t^\alpha}{\|\boldsymbol{a}\|_{\alpha^*}^\alpha} \Big) \Big)$$

**Lemma 4** (Moments for the Sum of Weibull Distributions (Corollary 1.2 in [49])). Let $X_1, X_2, \ldots, X_n$ be a sequence of independent symmetric random variables satisfying $\mathbb{P}(|Y_i| \geq t) = \exp(-t^\alpha)$, where $0 < \alpha < 1$. Then, for $p \geq 2$ and some constant $C(\alpha)$ which depends only on $\alpha$,

$$\left\| \sum_{i=1}^{n} a_i X_i \right\|_p \leq C(\alpha)(\sqrt{p}\|\boldsymbol{a}\|_2 + p^{1/\alpha}\|\boldsymbol{a}\|_\infty).$$

**Lemma 5** (Khinchin-Kahane Inequality (Theorem 1.3.1 in [50])). Let $\{a_i\}_{i=1}^n$ a finite non-random sequence, $\{\varepsilon_i\}_{i=1}^n$ be a sequence of independent Rademacher variables and $1 < p < q < \infty$. Then

$$\Big\| \sum_{i=1}^{n} \varepsilon_i a_i \Big\|_q \leq \Big( \frac{q-1}{p-1} \Big)^{1/2} \Big\| \sum_{i=1}^{n} \varepsilon_i a_i \Big\|_p.$$