[Reviews · NeurIPS 2019]

Reviewer 1



1. It is claimed that existing concentration-based confidence bounds are typically data-independent. This is true for the UCB1 algorithm, but there are other more sophisticated algorithms that exploit the full distribution in their confidence bound. For the example, for general distribution that have support in [0,1] the empirical KL-UCB of [Cappé et al., Kullback-Leibler Upper Confidence Bounds for Sequential Resource Allocation, 2013] used empirical likelihood to build confidence intervals, that are not at all of the form \bar y_n + data-independent terms. For Bernoulli bandits, more simple confidence intervals are proposed in the same papers (that extend to sub-Bernoulli, i.e. bounded, distributions). So I think it would be fair that those more sophisticated algorithms are also included in the comparison. Also, the analysis of UCB1 as proposed by [10] has been improved by several authors (using notably self-normalized deviation inequalities instead of union bounds) for sub-Gaussian distribution to show that the index \bar y_{n_k,t} + \sqrt{2\sigma^2log(t)/n_{k,t}} for sigma^2 sub-Gaussian distributions can be used (in place of the \sqrt{2log(t)/(n_{k,t})} originally proposed for 1/4-subGaussian). A fair comparison should include all the improvements from the literature. The experimental setup is also not clearly related to the theoretical guarantees that are obtained: while Theorem 2 hold for any fixed problem instance ("a" stochastic K-armed bandit, frequentist statement), it seems that the regret curves are obtained by averaging several runs on different randomly generated instances (Bayesian evaluation). Maybe I misunderstood something, but if the arms are fixed for good instead of being randomly generated in each run, one could as well provide their value. A Bayesian evaluation is interesting too to access the robustness of the algorithm on different problems, but given the nature of the theoretical results obtain, I think one or two "frequentist" regret curves are mandatory. In the linear bandit part, it seems the dimension d under which the experiments were run is not specified in Section 4.2. 2. The complexity of the algorithm is not discussed in details, it is just written in the introduction that it is "easy to implement". I should be acknowledged that it is significantly more complex that UCB1 for example. Indeed at each time step B bootstrap repetitions are needed to estimated the bootstrapped quantiles, and each of them require to drawn n_k random variables for each arm k (the values of w's). Also, this requires to store the past rewards obtained on all arms, which requires a lot a memory. This constraint is also needed for the empirical KL-UCB mentioned above, which is one more reason to compare the two algorithms that have similar complexity. From Theorem 2, I guess that the w's are Rademacher random variables, but it would be good to specify this in the statement of the algorithm. Bootstrapped UCB has two hyper-parameters, B and delta. Some insight on the parameter delta would be much appreciated. The tuning of the two parameters is never justified. We get that the larger B the better the algorithm and the more complex, but why B=200 specifically? Regarding delta, it is arbitrarily set to delta=0.1 in Section 4.1 and then to delta_t = 1/(1+t) for linear bandits "to be fair". I don't get why this is fair. Regarding the parameter alpha, I would like to mention that it is set to alpha=1/(t+1) in each round in the statement of Algorithm 1 (and I guess, the algorithm was implemented with this choice), however regret guarantees are only obtained by a fixed choice alpha = 1/T^2 where T is the full horizon. This discrepancy is annoying. 3. I checked the proofs of Theorem 2.2 and Theorem 3.2, which are the most important results of the paper. Note that the paper would be interesting even without the habillity to generalize to sub-Weibul distributions (not that actually, all experiments feature sub-Gaussian distributions, so there is not a strong case for this generalization. As such, it should be precised which function \phi is employed in the experiments. If beta=2 I would peferr to employ directly (2log(1/alpha)/n)^{1/2}), but I couldn't figure out what was done. I'm essentially OK with the proof of Theorem 3.2, though I didn't check too carefully the sub-Weibull tricks. I noted two typos in Equation (B.18) : u_1 should be \mu_1 twice. Also the notation \bar y_s is not super-precise as it sometimes refer to s i.i.d. samples from arm 1 or from arm k : I would introduce \bar y_{k,s} to avoid this aliasing. In the proof of Theorem 2.2, I have a hard time to understand where Equation (B.2) comes from, so I think detailed explanations are needed here. By definition I get that $\Pr(\bar y_n - \mu > q_\alpha(y_n - \mu) = \Pr_y(\Pr_w(1/n \sum_{i=1}^n w_i(y_i - \mu) > \bar y_n - \mu ) \leq alpha)$, but the formula in (B.2) seem to have inverted the integration over y and w in a way I don't understand. Also, the notation q_\alpha(z) for any vector z is not really defined, only q_\alpha(y_n - \mu) is defined in the paper: a more general notation should be introduced. The second problem I saw was on top on page 13, where some conditioning on event E is brutally removed: in the first inequality there should be a \bP(\bar y_n - \mu > q_\alpha(y_n - \mu) | E) + P() instead of the same thing without the conditioning. And the distribution of \bar y_n conditioned on the fact that y_n satisfy some condition is not necessarily the same as without the condition.

Reviewer 2



The paper is clear and well-written. I believe the main result Theorem 2.1 is novel. But I have the following concerns. (1) The results depends on the symmetry of the rewards. This is a huge assumption, which does not hold for many applications, including the Bernoulli bandits and many real-world problems with highly-skewed rewards. I do not take this as a downside of this paper, but this should be explicitly clarified in the abstract and the introduction to avoid overclaims. (2) The function \phi(y_n) is still needed in Theorem 2.2 as an exact concentration bound for \bar{y}_n - \mu. This is only possible in previously studied cases such as bounded rewards or Gaussian rewards. Admittedly this can also be extended to sub-gaussian or sub-Weibull rewards with known Orlicz norm, but the Orlicz norm is arguably never known in practice for potentially unbounded rewards. So from my point of view, the bootstrap UCB improves but does not extend the regime for the regret guarantees. Again I do not think this is a downside and I think the improvement is interesting, but this point should be made explicit at the beginning of the paper to avoid overclaims. (3) When comparing Bootstrap UCB with vanilla UCB, how do you set the alpha for both? Given the parameter \alpha, the confidence level for vanilla UCB is \alpha, while that in your theory (Theorem 2.1) is 2\alpha. For fair comparison, if you take \delta = 0.5, the equation (2.6) should be set as q_{\alpha / 4}(y_n - \bar{y}_n) + 2\log (8 / \alpha) / n. (4) In Theorem 3.2, \alpha_t is set to be 1/T^2 but in implementation it is set to be 1/(t+1). Would it be possible to analyze the latter as well? (5) The authors claim the sub-Weibull variables as "heavy-tailed" (e.g. the second line of Section 3.2). I do not think this is what people call "heavy-tailed". It usually means variables with only finite moments. The random variables with Weibull tail is light-tailed. (6) There are lots of missing references for the multiplier bootstrap. It can be dated back to Rubin (1981), which was called Bayesian bootstrap initially. Later it was studied and developed by Wu (1986), Liu (1988), Mason and Newton (1992), Rao and Zhao (1992), Mammen (1993), Chatterjee (1999), just to name a few. A relatively thorough literature review is important for a high-quality paper. References D. B. Rubin. The bayesian bootstrap. The annals of statistics, pages 130–134, 1981. C.-F. J. Wu. Jackknife, bootstrap and other resampling methods in regression analysis. the Annals of Statistics, 14(4):1261–1295, 1986. R. Y. Liu. Bootstrap procedures under some non-iid models. The Annals of Statistics, 16(4):1696–1708, 1988. Mason, David M., and Michael A. Newton. "A rank statistics approach to the consistency of a general bootstrap." The Annals of Statistics 20.3 (1992): 1611-1624. C. R. Rao and L. Zhao. Approximation to the distribution of M-estimates in linear models by randomly weighted bootstrap. Sankhya: The Indian Journal of Statistics, Series A, pages 323–331, 1992. E. Mammen. Bootstrap and wild bootstrap for high dimensional linear models. The annals of statistics, 21(1):255–285, 1993. S. B. Chatterjee. Generalised bootstrap techniques. PhD thesis, Indian Statistical Institute, Kolkata, 1999.

Reviewer 3



In this paper, the authors propose a novel point of view on a very well-known algorithm: UCB. Rather than using worst case concentration inequalities, which only exploit the tail information, the authors take advantage of the multiplayer bootstrap to provide a non-parametric data dependent UCB. The multiplayer bootstrap consists in approximating the quantile q_\alpha by reweighting the data with random multipliers independent of the data. Theorem 2.2 provides a significant result by controlling non-asymptotically the bootstrapped quantile. Indeed rather than using a worst case concentration inequality which leads to a data independent UCB, the control of the quantile allows to build a data dependent UCB. Bootstrapped UCB (algorithm 1) uses a Monte Carlo approach to approximate the bootstrapped quantile. The second significant analytical result is a concentration inequality for sub-Weibull Distribution, which is more general than sub-Gaussian distribution. Theorem 3.1 allows extending Bootstrapped UCB (and a lot of bandit algorithms) to sub-Weibull Distribution. Finally Theorem 3.3 states problem dependent and problem independent upper bounds of regret for Bootstrapped UCB. Experimental results show that Bootstrapped UCB outperforms UCB1, while it is more robust against a wrong prior than TS. This is a well-written paper which contains significant results for the bandit community. Nevertheless, I was disappointed by the fact that the control of the approximation of the bootstrapped quantile q^B_\alpha with respect to q_\alpha is not done. Is-it a big deal ? I understand that this control depends only on the parameter B, and hence on the computational cost. However sometimes, we cannot consider that the computational cost is not an issue, for instance for IoT. ______________________________________________________________ The authors answered to my concern. The obtained algorithm is still computationally expensive. However I think that the approach is original and could open research avenue for bandit community. I recommend acceptation.

[Author Response · NeurIPS 2019]

We would like to thank all reviewers for your valuable and detailed comments! We think we can fix all the issues raised
by the reviewers and add more experimental results in the final version of the paper. We hope you are satisfied with our
point-by-point responses and increase your scores.

**Response to Reviewer 1**

*Experiments.* Thank you for pointing out KL-UCB, a great and powerful algorithm for cases with bounded rewards. We
will compare with KL-UCB in detail, and have done comprehensive experiments with KL-UCB, using Prof. Olivier
Cappé's package online (due to the space limit, we can only report parts of them here). Our algorithm is comparable
with KL-UCB (Bernoulli, F1) and empirical KL-UCB (truncated normal, F2) under fixed problem instances suggested
in the package. Although KL-UCB is also data-dependent, our proposed method is from a very different non-parametric
perspective and uses different tools by bootstrap. In practice, resampling tends to be more efficient computationally,
without solving a convex optimization each round like empirical KL-UCB. Moreover, our method can work with
    unbounded rewards and we believe it is easier to generalize to structured bandits, e.g. linear bandit.

The UCB1 we use, defined as $\bar{y}_{n_{k,t}} + \sigma\sqrt{2\log(t)/n_{k,t}}$, exactly follows your suggestion (see E.1 in the appendix). So
we will change its name to vanilla UCB. In addition, we have added the frequentist regret curve (F3). Here, the regrets
of various algorithms are with respect to the instance gap ($\Delta$) and $\mu = (\Delta, 0, \dots, 0)$. In the linear bandit part, the
dimension is specified in the title of each figure.

*Complexity.* Indeed, our algorithm is more complex than vanilla UCB and requires more memories. The computational
complexity at step $t$ is $\widetilde{\mathcal{O}}(Bt) \leq \widetilde{\mathcal{O}}(BT)$. Comparing with vanilla UCB, the extra $Bt$ is due to resampling. We also
derive Theorem 0.1 below for MC quantile approximation error that provides us a theoretical guidance for the selection
of $B$. In practice, the choice of $B$ is seldom treated as a tuning parameter, but usually determined by the available
computational resource. To reduce the computational cost, we could use the idea of Bag of Little Bootstraps [2]. For
$\delta$, from (2.5), a smaller value of $\delta$ enables us to calculate the quantile at a closer level to $\alpha$ but will result in a larger
correction term. Since the correction term converges to 0 faster, we suggest a smaller $\delta$. In Section 4.2, the $\delta = 1/(1+t)$
is essentially the confidence level, rather than a hyper-parameter $\delta$ in (2.5). The choice of $1/T^2$ led to an easy analysis.
Using similar techniques in Chapter 8.2 of [1], we can derive a similar regret bound by setting $\alpha_t = 1/(t\log^\alpha(t))$ for
any $\alpha > 0$. We re-run the experiments and there is no significant change.

*Proof Clarifications.* Thank you for pointing out the typos and notations! We have revised them accordingly. For
Equation B.2, by the symmetric assumption of the reward, the distribution of $y_i - \mu$ is *exactly the same* as the distribution
of $w_i(y_i-\mu)$ for Rademacher r.v. $\{w_i\}$. This implies $\mathbb{P}_{\boldsymbol{y}}(\frac{1}{n}\sum_{i=1}^{n}(y_i-\mu) > q_\alpha(\boldsymbol{y}_n-\mu)) = \mathbb{P}_{\boldsymbol{y},\boldsymbol{w}}(\frac{1}{n}\sum_{i=1}^{n}w_i(y_i-\mu) >$
$q_\alpha((\boldsymbol{y}_n - \mu) \circ \boldsymbol{w}_n)) = \mathbb{E}_{\boldsymbol{w}}\mathbb{P}_{\boldsymbol{y}}\left(\frac{1}{n}\sum_{i=1}^{n}w_i(y_i - \mu) > q_\alpha((\boldsymbol{y}_n - \mu) \circ \boldsymbol{w}_n)\right)$. For the second one, instead of using
conditional event, we could use union event trick: $\mathbb{P}(A) = \mathbb{P}(A \cap B) + \mathbb{P}(A \cap B^c) \leq \mathbb{P}(A \cap B) + \mathbb{P}(B^c)$. By choosing
$A = \{\bar{y}_n - \mu > q_{\alpha(1-\delta)}(\boldsymbol{y}_n - \bar{y}_n) + (2\log(2/\alpha\delta)/n)^{1/2}\varphi(\boldsymbol{y}_n)\}$ and $B = \{\boldsymbol{y}_n \in \mathcal{E}\}$, we can reach the conclusion.

**Response to Reviewer 2**

(1)(2). We will explicitly clarify and mention them in the beginning. (3). Yes, you are right. (4). We can derive a similar
regret bound by setting $\alpha_t = 1/(t\log^\alpha(t))$ for any $\alpha > 0$. Please see our response for complexity part above. (5). We
will change the argument to heavier tail than sub-Gaussian/exponential. (6). Thanks for the detailed references. We will
add them in the introduction.

**Response to Reviewer 3**

Thank you for pointing the MC quantile approximation. We have derived the corresponding theorem for the control of
the approximation of the bootstrapped quantile.

**Theorem 0.1** (Monte Carlo Quantile Approximation)**.** Suppose the same conditions in Theorem 2.2 hold. We have
$\mathbb{P}_{\boldsymbol{y},\boldsymbol{w}}(\bar{y}_n - \mu > \widetilde{q}_\alpha^B + \sqrt{\log(2/\alpha\delta)/n}\varphi(\boldsymbol{y}_n)) \leq \alpha + \frac{\lfloor B\alpha \rfloor + 1}{B+1} \leq 2\alpha + \frac{1}{B+1}$, where $\widetilde{q}_\alpha^B$ is the Monte Carlo approximated
quantile defined in (D.1).

By replacing the true quantile $q_\alpha$ by a MC quantile $\widetilde{q}_\alpha^B$ based on $B$ i.i.d bootstrapped weights, we lose at most $1/(B+1)$
for the confidence level. The proof is similar to Theorem 2.2 except for the control of i.i.d approximation error.

[1]. Bandit Algorithms. Cambridge University Press (2019). [2]. The Big Data Bootstrap. ICML (2012).


[Meta-Review · NeurIPS 2019]

The reviewers updated their scores after the rebuttal and discussion. Congratulations on a nice paper that had a consensus on acceptance! The reviewers has a couple of outstanding concerns (like relating B,T) that I would like the authors to explicitly discuss (including potentially mentioning open problems) in the camera-ready version. All the best!